# RegexPSPACE: A Benchmark for Evaluating LLM Reasoning on PSPACE-complete Regex Problems

## Abstract

Large language models (LLMs) show strong performance across natural language processing (NLP), mathematical reasoning, and programming, and recent large reasoning models (LRMs) further emphasize explicit reasoning. Yet their computational limits—particularly spatial complexity constrained by finite context windows—remain poorly understood. While recent works often focus on problems within the NP complexity class, we push the boundary by introducing a novel benchmark grounded in two PSPACE-complete regular expression (regex) problems: equivalence decision (RegexEQ) and minimization (RegexMin). PSPACE-complete problems serve as a more rigorous standard for assessing computational capacity, as their solutions require massive search space exploration. We perform a double-exponential space exploration to construct a labeled dataset of over a million regex instances with a sound filtering process to build the benchmark. We conduct extensive evaluations on 6 LLMs and 5 LRMs of varying scales, revealing common failure patterns such as verbosity and repetition. With its well-defined structure and quantitative evaluation metrics, this work presents the first empirical investigation into the spatial computational limitations of LLMs and LRMs, offering a new framework for evaluating their advanced reasoning capabilities.

## 1 Introduction

The recent success of large language models (LLMs) has rapidly expanded their applications beyond traditional natural language processing (NLP) tasks to domains such as mathematical reasoning and programming. In particular, the emergence of large reasoning models (LRMs) has significantly advanced performance in areas of reasoning where conventional LLMs have struggled (Wei et al., 2022; Yao et al., 2023a;b). While the range of tasks that LLMs can handle continues to grow and their achievements are remarkable, a more fundamental question remains: *how much computational power do LLMs actually possess?* Theoretically, LLMs are often claimed to be Turing-complete (Pérez et al., 2019; 2021), yet such arguments rely on unrealistic assumptions, such as infinite context length, that are inconsistent with the practical constraints of models. Several recent studies (Fan et al., 2024a;b; Bampis et al., 2024) have attempted to measure LLMs' computational capabilities by employing benchmarks based on NP-hard problems. However, the actual limitations of LLMs are more closely tied to context length, which represents limited memory, and from this spatial perspective, analyses of their computational boundaries remain underexplored. Thus, empirically identifying the limits of LLMs' computational capacity under spatial constraints remains both a challenging and essential.

Fortunately, there is a class of problems, PSPACE, that requires polynomial space and is known to be harder than NP unless NP=PSPACE. One of the most well-known PSPACE-complete problems is determining a winning strategy in games such as chess or Go (Schaefer, 1978; Lichtenstein & Sipser, 1980; Storer, 1983), where the difficulty stems from the enormous search space. PSPACE-complete problems often require massive exploration since, under pure space constraints, it is possible to traverse the entire search space by imposing an ordering. Due to this characteristic, PSPACE-complete problems may admit local optima that differ from global optimum, which makes them particularly suitable for evaluating the reasoning ability of LLMs. However, there is a key obstacle that their intractability makes the construction of labeled datasets highly challenging. Moreover, these prob-

lems are difficult to evaluate quantitatively, as assessing the degree of error in incorrect outputs is often non-trivial.

In order to overcome these limitations, we focus on two well-known problems, regex minimization and equivalence decision. Regexes are a fundamental representation of regular languages— the simplest class in formal language theory— and are widely used in practice for tasks such as string search, pattern matching, and text preprocessing (Thompson, 1968a). Regex minimization and equivalence decision are practically important, as they contribute to producing more concise expressions and mitigating vulnerabilities such as Regex Denial of Service (ReDoS) (Li et al., 2022). Despite their practical relevance, the binary decision problem of testing equivalence between two regexes is PSPACE-complete, and minimizing a regex is also PSPACE-complete (Meyer & Stockmeyer, 1972). While these tasks also require massive exploration to establish ground truth labels, they offer naturally defined quantitative metrics, which enable the evaluation of model outputs and intermediate reasoning steps. In addition, due to the wide applicability of regex and their frequent appearance in code, all pretrained LLMs considered in our study have prior knowledge of regex. This reduces inequality across models stemming from differences in pretraining and makes these tasks particularly suitable for evaluation.

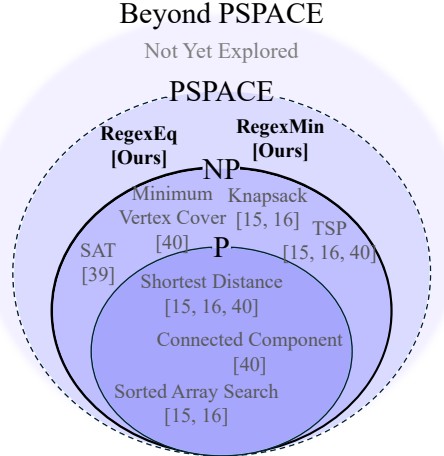

Figure 1: Overview diagram for the complexity class. In this work, we target PSPACE-complete problems, a class that has received relatively little exploration so far. The papers cited in the figure are as follows: Fan et al. (2024a) [15], Fan et al. (2024b) [16], Subramanian et al. (2025) [39], and Tang et al. (2025) [40]

Building on this well-defined foundation, we construct labeled datasets for the two regex-related PSPACE-complete tasks While pretrained LLMs demonstrate some level of understanding without additional training, they nevertheless struggle to solve the tasks. Trained models also show only limited generalization, performing well on regexes of lengths similar to those seen during training but failing to maintain performance on unseen lengths. This highlights the intrinsic difficulty of the tasks beyond mere exposure to training data. Based on these findings, we report discrepancies between the theoretical capacity and the practical ability of LLMs. We further argue for the necessity of evaluating LLM reasoning under spatial complexity constraints using challenging tasks of this nature. To this end, we introduce the RegexPSPACE benchmark, constructed through careful filtering of test sets, and present quantitative evaluation metrics along with results on 6 LLMs and 5 LRMs.

## 2 RELATED WORKS

One needs a basic background in the theory of computation, to fully understand our work, which is one of the fundamental areas of research in computer science. Naturally, this paper cannot cover all relevant material, but we summarize the necessary preliminaries related to our study in Appendix A. Because our work is the first to construct a dataset, a test set and a benchmark for the complexity class of PSPACE-complete problems, there is no direct prior work. Instead, we review prior analyses of the computational capabilities of LLMs, explorations of other complexity classes, and the real-world applications of regexes to motivate and justify our choice of tasks.

### 2.1 APPROACHES ANALYZING THE COMPUTATIONAL POWER OF LLMS

Since the advent of deep neural networks, one of the most persistent yet slow-progressing areas of research has been explaining neural models. Although neural models are undeniably computing machines that map inputs to outputs, their incorporation of nonlinearities for learning richer representations makes interpreting their behavior challenging. Nonetheless, there has been steady the-

oretical interest in evaluating their fundamental computational abilities, including several analyses of attention-based models. For instance, Pérez et al. (2021) argued that the Transformer's attention structure is capable of simulating a universal Turing machine, while Bhattamishra et al. (2020) further analyzed which component of the Transformer network is essential for the Turing-completeness of the network. More recent work (Dziri et al., 2023; Keles et al., 2023) attempted to analyze the computational complexity of RNNs, LSTMs, and Transformers by training them as classifiers over sets of strings. Despite these efforts, such studies often rely on theoretical assumptions or are limited to small-scale models rather than practically deployed LLMs. Thus, empirical investigations into the computational power of widely used LLMs remain insufficient and necessary.

## 2.2 Benchmarks for Exploring LLM Computational Power

Motivated by this need, several benchmarks have been proposed to evaluate the capabilities of LLMs. Fan et al. (2024a) introduced NPHardEval, a benchmark for assessing LLM performance on NP-hard problems, covering P, NP-complete, and NP-hard classes. As a follow-up, Fan et al. (2024b) proposed NPHardEval4V, an extension designed to evaluate multimodal LLMs using image-based tasks. Additionally, Qi et al. (2024) introduced SCYLLA, a benchmark incorporating problems with diverse time complexities, aiming to show that LLMs can generalize beyond rote memorization. However, these benchmarks predominantly focus on problems defined in terms of time complexity, while analyses from the perspective of space complexity remain scarce. Figure 1 summarizes the representative problems and related benchmarks across different complexity classes. As the table shows, while P and NP problems have been studied relatively extensively, no labeled datasets or benchmarks exist for PSPACE or beyond, primarily due to the intractability of labeling. Moreover, prior benchmarks often rely solely on accuracy-based metrics, making cross-task comparisons of difficulty and reasoning analysis limited. Our proposed RegexPSPACE benchmark takes a step further by evaluating the reasoning ability of LLMs on PSPACE-complete problems and providing task-specific evaluation metrics that enable diversified analysis.

## 2.3 Real-world Applications of Regexes and Relation to LLMs

Among the many PSPACE-complete problems, we chose regex-related tasks for several reasons. Most importantly, regexes have wide-ranging real-world applications, making the tasks inherently important. They are practically relevant in natural language processing (NLP), software engineering (SE), and programming languages (PL) (Davis et al., 2018; Shen et al., 2018; Li et al., 2022; Siddiq et al., 2024). In particular, finding concise and safe regex representations has always been essential in optimizing search engines (Thompson, 1968b). Because of their prevalence in search, preprocessing, and other applications, most LLMs have already been exposed to regexes during pretraining, enabling regex tasks to serve as natural benchmarks without requiring additional fine-tuning. From a task perspective, regex problems also provide advantages. Although equivalence decision is PSPACE-complete, existing libraries allow exponential-time testing for shorter lengths, which in turn enables rigorous evaluation metrics. Such metrics go beyond accuracy, allowing more detailed analyses of model outputs. Therefore, regex tasks are not only practically important and familiar to LLMs, but also well-suited as benchmarks due to their support for partial success evaluation through metrics like equivalence and length ratio.

## 3 Problem Descriptions

We target PSPACE-complete regex problems, specifically the tasks of regex equivalence decision and regex minimization. To the best of our knowledge, we are the first to propose these tasks as benchmarks. Therefore, in this section, we provide a clear description of the problems we address. A more formal definition using precise notation is provided in Appendix A.2.3 and A.2.4.

### 3.1 RegexEQ

Regex is a representation method for a set of strings that exhibits regular properties. A regex representing a given set of strings is not unique—multiple equivalent expressions may exist. Given two regular expressions, they are considered equivalent if they recognize the same set of strings. The regex equivalence decision task is to determine whether two given regexes recognize the same set of

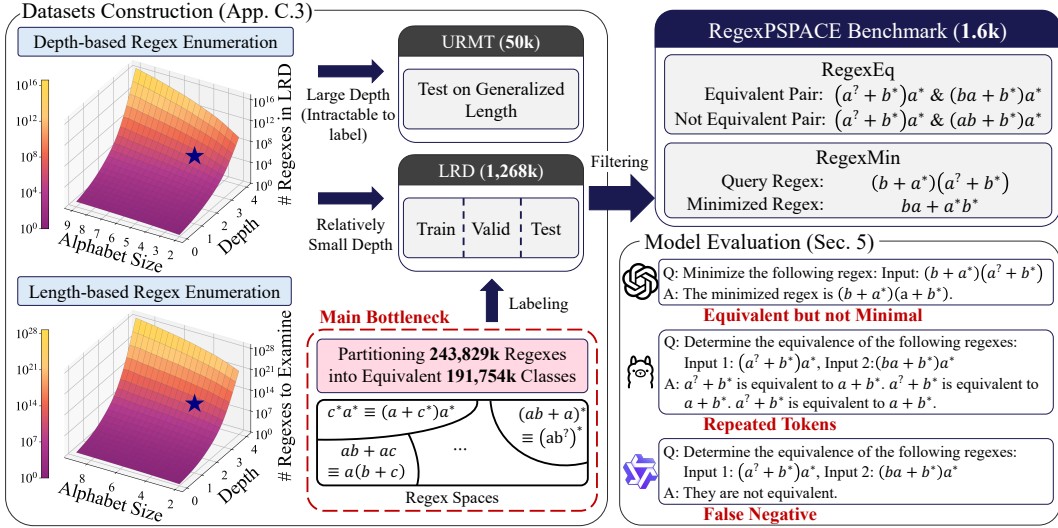

Figure 2: Overview of our work. We construct the labeled regex dataset (LRD) and the unlabeled regex minimization test set (URMT) and label LRD with the massive partitioning of regexes. The stars on the 3D graph visualize the number of regexes in the dataset and the number of regexes to examine for calculating minimality. We construct RegexPSPACE by filtering the test set of LRD and evaluate LLMs and LRMs on our benchmark.

strings, making it a fundamental binary decision problem. The PSPACE-completeness of RegexEq was reported by (Meyer & Stockmeyer, 1972).

## 3.2 REGEXMIN

As discussed earlier, a single regex can have multiple equivalent expressions. The goal of Regex Minimization is to find the shortest regex among all equivalent expressions. Unlike RegexEq, this task takes a single regex as input and aims to reduce its length as much as possible. Depending on the allowed operations and the definition of length, regex minimization has several variants. The most widely used definition measures length based on the number of operators and characters. We note that throughout this paper we adopt this definition, where concatenation—often omitted between characters—is also counted as an operator. Although equivalence decision and minimization belong to the same complexity class, regex minimization in practice requires substantially more computation and is therefore a harder problem. The PSPACE-completeness of Regex Minimization was proved by (Meyer & Stockmeyer, 1972).

## 4 DATASET CONSTRUCTION

We construct three resources for regex problems: a labeled dataset, an unlabeled dataset, and a benchmark. As noted earlier in Section 3, regex minimization is considerably more difficult in practice than regex equivalence decision. Therefore, we first construct a dataset for regex minimization containing more than one million regexes. Since an equivalence decision is inevitably required to determine the ground truth for minimization, sets of equivalent regexes are naturally obtained as a byproduct. Randomly splitting this large dataset into training, validating, and testing partitions yields our first dataset. We restrict the alphabet size and expression depth since double-exponential exploration is required to enumerate all regexes up to a given length and group equivalent ones into the same class.

While regexes can be arbitrarily long, enumerating all possible regexes is infeasible since they are infinite in practice. Therefore, in order to evaluate whether a trained model genuinely performs well on the task, it is necessary to examine its ability to generalize to instances with lengths unseen during

training. For this purpose, we construct a test dataset consisting of fifty thousands regexes longer than those used in the labeled dataset. However, since the labeled dataset already operates near the practical limit of expression depth, obtaining ground truth for longer regexes is nearly impossible. Thus, we construct an unlabeled test dataset for assessing generalization. Analysis on computational effort to construct the dataset is given in Appendix E. This dataset is sampled in a manner similar to the labeled dataset but does not include ground truth minimal regexes or equivalence labels, and can therefore only be used for the minimization task. As it lacks ground truth labels, model performance is evaluated indirectly through equivalence preservation and reduction ratios.

Our preliminary experiments on RegexMin in Appendix F reveal that regex minimization is both a challenging and intuitively aligned evaluation task, underscoring its value as a benchmark for reasoning performance. Based on this, we refine the test set of the labeled dataset, augment it with another PSPACE-complete task of regex equivalence, and curate 1,685 non-trivial regex problems through careful filtering to construct the Regex Problem Benchmark. Details of the construction precess are provided in Appendix B.

## 5 EVALUATION OF LLMS ON REGEXPSPACE

We evaluate a diverse set of pretrained LLMs and LRMs of varying sizes and from different providers in order to examine LLM performance on RegexPSPACE and identify general trends. Specifically, we conduct evaluations on six LLMs, including Qwen, Qwen-Coder, Llama, and Phi-4, as well as six LRMs, including DeepSeek-R1 (DS-R1), Phi-4, and gpt-oss. The models are selected to investigate the effects of size, reasoning capability, and specialization for code. For each task, we employ manually crafted prompts and conduct experiments in both zero-shot and five-shot settings. Detailed information about the models, prompts, and experimental hyperparameters is provided in Appendix G.

### 5.1 METRICS

Our evaluation relies on carefully chosen metrics that quantify the validity of model outputs. For evaluating RegexMin, we report three metrics: minimality, equivalence, and length ratio. For evaluating RegexEq, we report accuracy and F1-score. Details of each metric and their formal definitions are provided in Appendix G.2.

**Metrics for RegexMin**   Minimality measures the proportion of responses that are equivalent to the query regex while also having minimal length. Equivalence measures the proportion of responses that are equivalent to the query regex but not necessarily minimal. Minimality is always less than or equal to equivalence, and higher values of both metrics indicate better performance. Length ratio, defined by (Kahrs & Runciman, 2022), is the average ratio of the length of the output regex to that of the original regex. We treat responses that are not equivalent to the query regex as non-reductions. Unlike the other metrics, a smaller length ratio indicates better performance. The best achievable performance on RegexMin is 0.7270.

**Metrics for RegexEq**   Accuracy measures the proportion of correctly predicted labels indicating whether a given pair of regexes is equivalent. Since binary classification with consistent outputs of either True or False can trivially achieve 50% accuracy, we additionally report F1-score, which considers both precision and recall. As the evaluation is performed via prompting, models may fail to output a valid label. Therefore, F1-score is computed only over outputs where the model successfully produced either a True or False label. It should not be interpreted in isolation but rather together with the proportion of invalid outputs.

### 5.2 OVERALL PERFORMANCE

**Task Dependency**   The experimental results are summarized in Table 1. Overall, the models struggle significantly on the minimization task. From the minimization perspective, most models not only fail to produce shorter regexes but also struggle to output even equivalent ones, with the proportion of equivalent outputs falling below 50% for the majority of models. This highlights the inherent difficulty of the minimization task for current models. In terms of length ratio, while the best achievable

Table 1: Our main evaluation results on RegexPSPACE. We evaluate 6 LLMs and 5 LRMs across diverse model families and sizes. We report zero-shot and 5-shot prompting results on both RegexMin and RegexEq. For clarity, the best-performing LLMs and LRMs for each metric are highlighted in bold. We additionally report fail rates, since the F1-score is computed only from correctly parsed answer. For the length ratio, the best achievable performance on RegexPSPACE is 72.70%.

| Model | Size | Shot | RegexMin | | | RegexEq | | |
|---|---|---|---|---|---|---|---|---|
| | | | Min. (↑) | Equi. (↑) | Ratio (↓) | Acc. (↑) | F1 (↑) | Fail (↓) |
| Qwen2.5 | 7B | Zero | 20.00 | 25.04 | 91.67 | 62.08 | 59.46 | 1.66 |
| | | Five | 10.98 | 14.90 | 94.71 | 64.07 | 63.15 | 0.86 |
| Qwen2.5-Coder | 7B | Zero | 14.18 | 21.90 | 94.05 | 58.72 | 51.98 | 0.53 |
| | | Five | 17.09 | 40.42 | 92.94 | 65.85 | **72.90** | **0.00** |
| Llama-3.1 | 8B | Zero | 2.31 | 3.98 | 98.60 | 30.45 | 42.81 | 46.08 |
| | | Five | 3.50 | 4.21 | 97.96 | 36.11 | 42.62 | 34.81 |
| Phi-4 | 14B | Zero | 23.32 | 25.34 | 90.56 | 57.06 | 32.79 | 2.55 |
| | | Five | 24.63 | 26.35 | 90.64 | 57.00 | 32.41 | 0.42 |
| Qwen2.5 | 14B | Zero | 27.24 | 32.40 | 89.68 | 63.53 | 50.63 | 0.18 |
| | | Five | 20.30 | 21.90 | 91.26 | **69.11** | 65.74 | 0.09 |
| Qwen2.5-Coder | 14B | Zero | 28.72 | **42.08** | 89.15 | 55.07 | 23.33 | 0.06 |
| | | Five | **31.93** | 40.83 | **87.82** | 58.69 | 38.40 | 0.18 |
| DS-R1-Qwen | 7B | Zero | 19.23 | 28.78 | 92.57 | 59.41 | 48.51 | 3.74 |
| | | Five | 17.74 | 23.09 | 92.87 | 58.07 | 51.02 | 6.26 |
| DS-R1-Llama | 8B | Zero | 1.19 | 3.09 | 99.57 | 24.33 | 14.07 | 55.73 |
| | | Five | 1.19 | 1.96 | 99.27 | 29.70 | 12.00 | 45.04 |
| DS-R1-Qwen | 14B | Zero | 30.92 | 35.55 | 88.46 | 61.87 | 74.66 | 23.20 |
| | | Five | 25.99 | 30.09 | 89.53 | 66.41 | 81.06 | 21.04 |
| Phi-4-reasoning | 15B | Zero | 34.84 | 47.00 | 87.35 | 45.46 | 73.61 | 42.82 |
| | | Five | 40.83 | 51.45 | 86.06 | 51.84 | 69.69 | 33.26 |
| gpt-oss-low | 20B | Zero | 67.36 | 84.45 | 78.48 | 84.96 | 82.85 | **0.03** |
| | | Five | **69.02** | **87.24** | **77.96** | **87.98** | 86.75 | **0.03** |
| gpt-oss-high | 20B | Zero | 4.57 | 4.69 | 97.71 | 62.94 | 95.26 | 34.66 |
| | | Five | 9.02 | 9.55 | 95.33 | 63.15 | **96.41** | 34.90 |

performance is 72.70%, most models achieve only around a 10% reduction. In contrast, performance is noticeably better on the equivalence decision task, even for models that fail on minimization. We attribute this difference to the nature of the tasks: equivalence requires only a binary decision of True or False, whereas minimization requires producing a valid regex that is not only equivalent to the query but also shorter in length. Furthermore, the RegexEq task is balanced between equivalent and non-equivalent pairs, meaning that a model consistently outputting only True or only False can trivially achieve 50% accuracy. In order to address this, we additionally report F1-scores based only on responses where the model successfully output either True or False.

**Model Size Dependency** From the perspective of model size, models with 14–15B parameters or larger generally outperform those with 7–8B parameters. This aligns with common intuition, as larger models are better equipped to handle long-term massive exploration. Similarly, reasoning models tend to perform better on minimization tasks compared to non-reasoning models, with reasoning models above 14B consistently outperforming non-reasoning ones. However, in equivalence tasks, 7-8B models and reasoning models do not show a clear advantage over non-reasoning models in terms of accuracy. Indeed, reasoning models are often observed to generate long sequences of repetitive phrases during the thinking phase or fail to complete their answers within the token limits. When F1-score is computed only over valid True/False outputs, reasoning models larger than 14B show a clear tendency to outperform both smaller models and non-reasoning ones.

Furthermore, we conduct experiments on several non-reasoning 30B models. Across all of those 30B models, we consistently observe excessively long responses that fail to terminate properly. A detailed analysis of these results is provided in Appendix H.2.

**Few-shot Examples Dependency**   RegexEq generally benefits from few-shot examples, with performance improving when examples are provided. RegexMin does not always exhibit such gains in minimality, but most non-reasoning models show a higher proportion of valid regex outputs with few-shot prompting, suggesting improvements in formatting and adherence to syntactic rules. We attribute the discrepancy, that few-shot examples help with formatting but not with minimization, to the fact that heuristics useful for specific examples do not generalize well, a limitation rooted in the PSPACE-complete nature of the problem. This contrast indicates that few-shot prompting primarily reinforces surface-level regularities rather than enabling deeper optimization, allowing models to mimic valid structures while still struggling with the combinatorial reasoning required for true minimization.

**Model Dependency**   The tendencies described above vary across models. Within the Qwen family, coder-specialized models consistently outperform their non-coder counterparts, likely because their ability to handle code naturally transfers to regex. DeepSeek models, Llama models, and the gpt-oss model in high reasoning mode frequently generate repetitive sequences during the production of either thinking or answer tokens. Such repetitions directly hinder response generation and therefore degrade performance. We further analyze how many failed regex generations could be attributed to repetition in the following section. Failures caused by repetition are more prevalent in RegexMin than in RegexEq. We attribute this difference to the nature of minimization, which requires repeatedly exploring similar forms of regex. During this process, models can become trapped in a repetitive probability distribution. Although sampling-based decoding could potentially mitigate this issue compared to greedy decoding, such probabilistic escapes still require large amounts of decoding and therefore do not constitute a fundamental solution. The best-performing model overall is gpt-oss, which achieves the strongest results in low reasoning mode. However, in high reasoning mode, it exhibits frequent repetition issues similar to DeepSeek, often failing to provide correct answers and generating until hitting the token limit.

## 5.3   Failure case analysis

In the Main Results, we observed that LLMs fail on tasks for a variety of distinct reasons. We categorize and examine the failure cases to more closely analyze the main challenges faced by LLMs and to validate our interpretation. For RegexMin, we classify each case into one success type and five failure types: (1) cases where the model successfully outputs a minimal regex, (2) cases where the output is not minimal but equivalent, (3) cases where the output is a valid expression but not equivalent, (4) cases where generation naturally stopped but produced an invalid expression, (5) cases where failure occurred due to repetition, and (6) cases where the model stopped after reaching the token limit without repetition. The failure cases are ordered by increasing severity, except for the last one. The final case—stopping due to the token limit without repetition—indicates that the model was still in the middle of generation and that the token budget we imposed was insufficient to produce a complete answer. Therefore, this case differs in both cause and nature from the others, and its severity is not directly comparable. However, if a model consistently requires more tokens than others within the same budget, it implies that the model demands more extensive exploration, which signals inefficiency rather than a positive property. For RegexEq, we report the proportions of True Positives (TP), True Negatives (TN), False Positives (FP), False Negatives (FN), cases where generation naturally stopped but yielded an invalid expression, cases where failure occurred due to repetition, and cases where the model stopped after reaching the token limit without repetition. Due to space constraints, in this section we only visualize the zero-shot results, while the full analysis is provided in Appendix H.1.

As shown in both Figure 3 and Figure 4, repetition is particularly pronounced in Llama, DeepSeek, and the high reasoning mode of gpt-oss. In addition, gpt-oss and phi-4-reasoning frequently terminate generation prematurely due to reaching the token limit. With respect to RegexMin, we also observe that non-reasoning models, particularly DeepSeek, produce a considerable number of invalid outputs. These outputs often arise from the use of practical regex notation, mathematical syntax, or other forms that cannot be easily parsed by rule-based mechanisms. Models without reasoning capabilities, including Qwen, frequently succeed in producing syntactically valid outputs. However, these are rarely equivalent to the target regex or minimal in form.

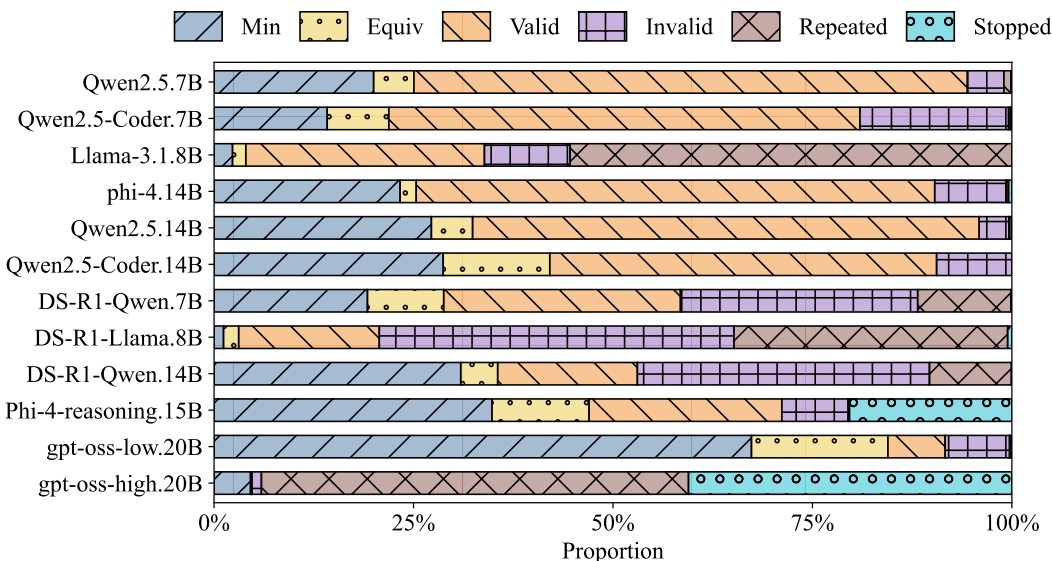

Figure 3: Case analysis bar chart of the zero-shot prompting results on RegexMin. The outcomes are categorized into Minimality, Not minimal but equivalent, Not equivalent but valid, Invalid but completed answers, Repetition, and Incomplete outputs.

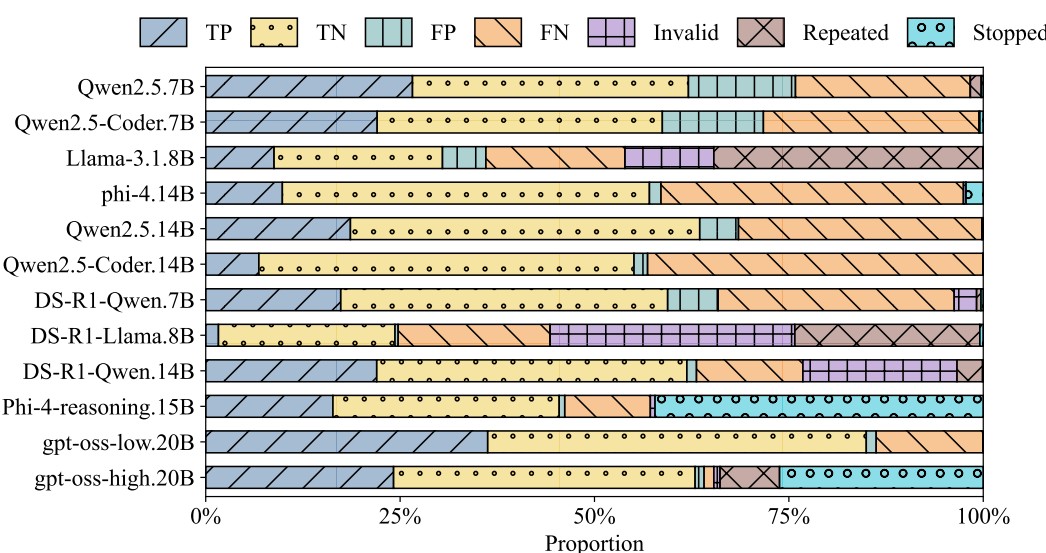

Figure 4: Case analysis bar chart of the zero-shot prompting results on RegexEq. The outcomes are categorized into the components of the confusion matrix, together with Invalid but completed answers, Repetition, and Incomplete outputs.

For RegexEq, the patterns of repetition and invalid outputs across models are broadly similar to those observed in RegexMin. Overall, models tend to output negative decisions more frequently than positive ones, revealing a systematic bias toward judging two regexes as non-equivalent. In contrast to its behavior on RegexMin, where gpt-oss often failed by repetition or incomplete outputs, its performance on RegexEq shows a higher proportion of correct answers and a substantial reduction in repetition. The incidence of invalid outputs is also markedly lower compared to RegexMin, confirming once again that equivalence checking is a relatively easier task for models to handle.

## 5.4 CORRELATION AMONG PERFORMANCE, ANSWER LENGTH, AND INPUT SIZE

One of our objectives in studying PSPACE-complete problems was to examine how problem difficulty, represented by input length, influences both the number of reasoning answer tokens and overall performance. Since performance variation across models was more pronounced in RegexMin, we conducted our analysis on this task. As shown in Figure 12, we report model performance with respect to the input query length, alongside the corresponding output token length. In general, longer inputs make the task more challenging, and solving them effectively tends to require generating more answer tokens. Indeed, we observed that models achieved higher performance on shorter regexes, with a slight upward trend in the average number of generated tokens as the input regex length increased. This phenomenon was particularly noticeable in non-reasoning models such as Qwen2.5. We attribute this to the fact that reasoning models often separate the thinking process into dedicated thinking tokens, enabling them to "reason" in advance while still producing relatively short answers.

## 6 CONCLUSION

We introduced RegexPSPACE, the first benchmark to evaluate LLMs and LRMs on PSPACE-complete regex problems. Our results show a clear gap between models' theoretical capacity and their practical performance. Specifically, models struggle with minimization, often failing to produce equivalent or shorter regexes, while performing relatively better on equivalence tasks. We also identified common failure patterns, such as verbosity, repetition, and premature stopping. These findings highlight the intrinsic difficulty of PSPACE-complete reasoning and the current limits of LLMs under spatial constraints. RegexPSPACE thus provides not only a rigorous evaluation framework but also a foundation for future research aimed at advancing reasoning capabilities beyond NP-level challenges.

**Limitations** One limitation of our dataset construction is the restriction to relatively small alphabet sizes and expression depths, imposed by practical constraints. As the alphabet or depth increases, the search space grows double-exponentially, making enumeration and minimization significantly harder. While such restrictions limit the discovery of more complex minimization rules and finer-grained reasoning abilities of LLMs, some form of limitation is unavoidable, given the infinite nature of regex. Through the computational cost analysis in Appendix E, we emphasize that our exploration is already massive and close to the feasible boundary. Extending to longer regex remains nearly intractable, yet it also represents a meaningful direction for future research.

Another limitation is that we focuse exclusively on regex-related PSPACE-complete problems. Although there are many PSPACE-complete problems, performance on regex tasks may not directly translate to other domains. Exploring additional PSPACE problems therefore remains an important open challenge. Nevertheless, the problems we target belong to the PSPACE-complete class and any other PSPACE problem can be reduced to them. Moreover, we introduce tasks that address complexity classes which have been scarcely explored, providing a valuable starting point for examining LLMs' reasoning ability from a spatial perspective. In addition, regex-related problems, particularly minimization, have the advantage of enabling quantitative evaluation even for partial success, through metrics such as equivalence and length ratio.

**Future Works** We propose several directions for future work to further extend the explored boundaries of LLMs' computational complexity. First, the most immediate step is to improve the performance of LLMs and LRMs on the RegexPSPACE benchmark. Most LLMs demonstrated poor performance on our benchmark, frequently repeating tokens during long reasoning processes or failing to complete responses within the token limit, both of which highlight current limitations in reasoning. Overcoming these issues and enhancing performance would represent an important step toward advancing the computational complexity capabilities of LLMs.

Beyond this, expanding to other PSPACE-complete domains also remains a significant challenge. As noted in the limitation section, RegexPSPACE is a valuable starting point, but broader exploration of PSPACE-complete problems is essential. Such tasks can serve as rigorous standards for measuring the reasoning capabilities of LLMs. Future research should further analyze and experiment with massive exploration under limited memory constraints, and investigate strategies for overcoming these barriers.

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

## A  PRELIMINARY

We cover the basics of formal language theory and computational complexity to aid in understanding our paper. For a more detailed explanation, refer to the book on the theory of computation by Sipser (1997).

### A.1  FORMAL LANGUAGES AND CHOMSKY HIERARCHY

An alphabet $\Sigma$ is a finite set of symbols, and a character $\sigma$ is an element of an alphabet. A string $w \in \Sigma^*$ is defined as a sequence of characters, and a formal language $L$ is defined as a set of strings over an alphabet. The Chomsky hierarchy classifies formal languages into four levels based on their complexity and the type of automaton required to recognize them as illustrated in Table 2.

Table 2: The Chomsky Hierarchy of Formal Languages.

| Type | Language | Recognizing Automaton |
|---|---|---|
| Type-0 | Recursively Enumerable | Turing Machine (TM) |
| Type-1 | Context-Sensitive | Linear-Bounded Automaton (LBA) |
| Type-2 | Context-Free | Pushdown Automaton (PDA) |
| Type-3 | Regular | Finite Automaton (FA) |

## A.2 REGULAR LANGUAGES

A language is called a *regular language* if it can be recognized by either a regular expression or a finite automaton. We present details of a regular expression and a finite automaton in Sections A.2.1 and A.2.2, respectively.

### A.2.1 REGULAR EXPRESSION

A regular expression (regex) defines a regular language, which forms the most basic set of formal languages. From a regex, a set of strings is defined, and this set is called its regular language. While there are several variants, the classical definition of a regex involves a finite alphabet $\Sigma$ and use two binary operations, concatenation and union, and one unary operation, the Kleene star. Following the problem setting of the most recent regex simplification work (Kahrs & Runciman, 2022), we adopt a definition that includes an additional unary operation, the option. We follows the notations for regex used in Gruber & Gulan (2010). The form of expressions is as follows: $\varepsilon, s, x + y, x \cdot y, x^*, x^?$ where $s \in \Sigma$, and $x, y$ are any finite expressions. For an expression $x$, the corresponding language $L(x)$ is defined as follows:

- $\emptyset = \{\}$ (An empty language)
- $L(\varepsilon) = \{\varepsilon\}$ (An empty string)
- $L(s) = \{s\} \ \forall s \in \Sigma$ (A character)
- $L(x + y) = L(x) \cup L(y)$ (Union)
- $L(x \cdot y) = \{vw | \forall v \in L(x), \forall w \in L(y)\}$ (Concatenation)
- $L(x^?) = L(x) \cup L(\varepsilon)$ (Option)
- $L(x^*) = \{w = x_1 \ldots x_n | \forall n \in N_0, \forall x_1, \ldots, x_n \in L(x)\}$ (Kleene Star)

By definition, $\Sigma^*$ denotes the set of all strings on the alphabet $\Sigma$ including $\varepsilon$. For convenience, we denote the space of all regexes over a fixed alphabet $\Sigma$ as $X(\Sigma)$. Since theoretical regexes are defined using unary and binary operations, every regex has a binary expression tree notation, where all internal nodes are operations.

### A.2.2 FINITE AUTOMATON

A finite-automaton (FA) $A = (Q, \Sigma, \delta, i, F)$ consists of a set $Q$ of states, and alphabet $\Sigma$, a transition function $\delta : Q \times \Sigma \to Q$, a start state $i \in Q$, and a set $F \subseteq Q$ of final states. We refer to an FA as a deterministic FA (DFA) when its transition function $\delta$ specifies exactly one next state for each combination of a state and an input symbol. We refer to an FA as a nondeterministic FA (NFA) otherwise. An FA $A$ *accepts* an input $w \in \Sigma^*$ if a sequence of states $q_0, q_1, \ldots, q_n \in Q$ exists, where

1. $q_0$ is a start state $i$,
2. $\delta(q_i, w_{i+1}) = q_{i+1}$, for $i = 0, \ldots, n - 1$, and
3. $q_n$ an element of a set $F$ of final states.

### A.2.3 EQUIVALENCE DECISION

A regex representing a regular language is not unique, and multiple equivalent expressions may exist for the same language. Two regexes are defined as equivalent if they recognize the same set

of strings, i.e., the same language. Formally, the equivalence between two regexes $x, y \in X(\Sigma)$ is denoted as $x \equiv y$. In this notation, the regex equivalence decision (RegexEq) is defined as follows: Given an alphabet $\Sigma$ and two regexes $x, y \in X(\Sigma)$, determine whether $x \equiv y$. The most naive approach for checking regex equivalence is to convert a regular expression into a nondeterministic finite automaton (NFA), transform it into a deterministic finite automaton (DFA), minimize the DFA, and then compare the minimal DFAs for equivalence. However, the NFA-to-DFA conversion step (subset construction) requires exponential space, leading to an EXPSPACE procedure. In fact, it is known that equivalence can be verified within PSPACE by simulating the DFA lazily with out explicitly constructing it, and more precisely, the problem has been shown to be PSPACE-complete (Meyer & Stockmeyer, 1972).

### A.2.4 MINIMIZATION

In order to formally define regex minimization, we must first specify a measure of regex length. Although several definitions exist, in this work we follow prior work (Ellul et al., 2004; Lee & Shallit, 2005; Gruber & Gulan, 2010; Kahrs & Runciman, 2022) and adopt the length defined by the number of nodes in the expression tree, which is the most widely used definition. Accordingly, minimal regexes are defined as those with the smallest tree length among all equivalent regexes. We use the term length of a regex interchangeably with its tree length and denote the tree length of a regex $r$ by $|r|_T$. For an alphabet $\Sigma$ and a regex $r \in X(\Sigma)$, we denote the set of regexes equivalent to $r$ as $X_r(\Sigma) = \{x \in X(\Sigma) | x \equiv r\}$, and the subset of $X_r(\Sigma)$ with minimal regexes is denoted as $X_r^\star(\Sigma) = \{x \in X_r(\Sigma) | \forall y \in X_r(\Sigma), |x|_T \leq |y|_T\}$. Given a regex $r \in X_r(\Sigma)$, the regex minimization (RegexMin) is to find a minimal equivalent regex $x \in X_r^\star(\Sigma)$. In practice, regex minimization requires traversing all regexes of length no greater than that of the input regex and checking equivalence. Since equivalence decision is already PSPACE-complete, and traversing shorter regexes with some ordering can also be resolved within a polynomial bound, regex minimization is likewise PSPACE-complete. From a complexity-theoretic perspective, regex minimization and regex equivalence belong to the same class. However, in practice, regex minimization typically requires far more extensive computation and is therefore considered the more challenging task.

### A.3 TURING MACHINE

A Turing machine (TM) $M = (Q, \Sigma, \Gamma, \delta, i, acc, rej)$ consists of a set $Q$ of states, an input alphabet $\Sigma$, a tape alphabet $\Gamma$, a transition function $\delta : Q \times \Gamma \to Q \times \Gamma \times \{L, R\}$, an initial state $q_0 \in Q$, an accepting state $acc \in Q$, and a rejecting state $rej \in Q$. A TM is *deterministic* when for every combination of a state and a tape symbol, its transition function $\delta$ specifies exactly one action. A TM is *nondeterministic* otherwise.

A *configuration* $c$ of a TM $M$ is a string in $\Gamma^* Q \Gamma^*$. A configuration $c_1$ *yields* a configuration $c_2$ if the TM $M$ can go from $c_1$ to $c_2$ in a single step. A TM $M$ accepts an input $w \in \Sigma^*$ if a sequence of configurations $c_1, c_2, \ldots, c_k \in \Gamma^* Q \Gamma^*$ exists, where

1. $c_1$ is a start configuration on input $w$, which is $q_0 w$,

2. each $c_i$ yields $c_{i+1}$, and

3. $c_k$ is an accepting configuration.

### A.4 COMPLEXITY OF PROBLEMS

The complexity of a problem is formally defined using the concept of a Turing machine, which generalizes all computing devices. A Turing machine is a computational model consisting of a single tape of infinite length, a head that operates on the tape, a finite set of states, and a transition function between these states. Modern computer architectures with unbounded memory are equivalent in computational power to a Turing machine. If the computation of a Turing machine is deterministic (resp. nondeterministic), we refer to it as a deterministic (resp. nondeterministic) Turing machine. The NP class is the set of problems that can be solved in polynomial time by a nondeterministic Turing machine. The PSPACE class is the set of problems that can be solved by a Turing machine whose tape usage is bounded by a polynomial function of the input length. It is a well-known fact that NP is a subset of PSPACE, which implies that PSPACE is at least as hard as NP unless NP=PSPACE (Sipser, 1997). Furthermore, the notions of completeness and hardness are

defined via reductions. For a given problem, if any NP (resp. PSPACE) problem can be reduced to it, then we call the problem NP-hard (resp. PSPACE-hard). If a problem is NP-hard (resp. PSPACE-hard) and NP (PSPACE), then it is NP-complete (PSPACE-complete). Since PSPACE is believed to be harder than NP, it follows that PSPACE-complete problems are NP-complete.

As shown in Figure 1 in Section 1, most of the existing labeled datasets address problems in P or NP, and the performance of LLMs on problems, and the performance of LLMs on problems believed to be harder—those in PSPACE or beyond—remains largely underexplored. Several studies have argued that the attention mechanism of LLMs possesses a Turing-complete nature, which lies far beyond the scope of the diagram. However, experimentally validating whether LLMs actually exhibit such capability is a different challenge. From the perspective of analyzing the computational power of LLMs, the context window is often regarded as analogous to the tape (memory) of a Turing machine. Under this abstraction, LLMs can be viewed as Non-erasing Turing Machines (NETMs), since they cannot overwrite previously generated tokens. Although NETMs have the same computational power as general Turing machines, simulating a standard Turing machine requires space bounded by the number of edits. This implies that simulating PSPACE-complete problems sequentially with LLMs would require EXPSPACE, which highlights the fundamental challenge of enabling LLMs to perform massive exploration.

In fact, there have already been attempts to approximate PSPACE-complete problems (Kautz & Selman, 1992; Kocsis & Szepesvári, 2006; Mahmood & Virtema, 2023). Notably, the generalization of perfect-information games such as Go and chess leads to PSPACE-complete complexity, and certain optimization problems, such as partial searchlight scheduling or true quantified boolean formula (TQBF), are also PSPACE-complete. However, such approaches neither provide access to the ground truth solutions nor offer a reliable way to determine whether a proposed solution is correct. In contrast, our proposed dataset and benchmark are the first labeled PSPACE-complete dataset constructed through massive computation. We further guarantee the correctness of our dataset through proofs and introduce quantitative metrics that enable rigorous evaluation.

### A.4.1 COMPLEXITY CLASSES

Complexity classes are defined over both time and space, which are essential for classifying the complexity of computational problems. The time complexity class $\text{TIME}(t(n))$ is the collection of all languages that are decidable by an $O(t(n))$ time deterministic TM over $t : \mathcal{N} \to \mathcal{R}^+$. The space complexity class $\text{SPACE}(f(n))$ is the collection of all languages that are decidable by an $O(f(n))$ space deterministic TM over $f : \mathcal{N} \to \mathcal{R}^+$.

**P** The class **P** consists of all decision problems that can be solved by a deterministic TM in polynomial time. That is,

$$P = \bigcup_k \text{TIME}(n^k).$$

**NP** The class **NP** consists of languages that have a polynomial-time verifier. A verifier for a language $L$ is an algorithm $A$, where

$$L = \{w \mid A \text{ accepts } \langle w, c \rangle \text{ some string } c\}.$$

The verifier $A$ for $L$ is a *polynomial-time verifier* if it runs in polynomial time in the length of $w \in L$.

**PSPACE** The class **PSPACE** consists of all decision problems that are decidable in polynomial space on a deterministic TM. In other words,

$$\text{PSPACE} = \bigcup_k \text{SPACE}(n^k).$$

**Beyond PSPACE** There exist complexity classes for languages considered more complex than those in PSPACE. Two notable examples are **EXPTIME** and **EXPSPACE**. EXPTIME contains problems solvable by a deterministic TM in exponential time, while EXPSPACE contains problems solvable in exponential space. The following hierarchy holds:

$$\text{P} \subseteq \text{NP} \subseteq \text{PSPACE} \subseteq \text{EXPTIME} \subseteq \text{EXPSPACE}.$$

### A.4.2 Reduction between Problems

A *reduction* is a procedure for converting one problem into another. A *polynomial-time reduction* from a problem $A$ to a problem $B$ is an algorithm that converts $A$ into $B$ in polynomial time. If such a reduction exists, we write $A \leq_P B$. This implies that $B$ is at least as hard as $A$ because an efficient solution for $B$ would provide an efficient solution for $A$.

### A.4.3 Complete Classes

A problem $B$ is **C-complete** for a complexity class C when the following conditions hold:

1. $B$ is in C, and
2. every $A$ in C is polynomial-time reducible to $B$.

Problems such boolean satisfiability problem (SAT), blah blah blah, are NP-complete, and the quantified boolean formula (QBF), blah blah blah, are PSPACE-complete. RegexEq and RegexMin from Sections A.2.3 and A.2.4 also are PSPACE-complete.

### A.4.4 Hard Classes

A problem $B$ is **C-hard** if every problem $A$ in a complexity class C is polynomial-time reducible to $B$. Unlike a C-complete problem, a C-hard problems is not required to be in the class C. For instance, a problem can be NP-hard but not be in NP.

## B Detailed Dataset Construction

We introduce a dataset for regex minimization, structured into two components: the Regex Minimization Corpus (LRD) and the Extended Regex Minimization Benchmark (URMT). LRD is a labeled dataset containing regexes and their minimal equivalents. It is partitioned into train, validation, and test sets. The minimality of a regex is verified by comparing it with all regexes of smaller lengths. This requirement causes a practical upper bound to the size of the regex minimization dataset.

Regex minimization, however, is not constrained by length in principle. Evaluating the ability of a model to generalize beyond observed lengths is therefore necessary. In order to address this, we construct URMT as a challenging out-of-distribution (OOD) benchmark, containing regexes longer than those in LRD. URMT is unlabeled and consists only of a test set. Dataset construction follows two main stages: 1) a bottom-up approach to generate regexes for the dataset and 2) the computation of minimal tree lengths. The size of alphabet $\Sigma$ is fixed as four, represented as $\{a, b, c, d\}$.

### B.1 LRD Construction

LRD is constructed using a bottom-up approach, leveraging the fact that a regex can be represented as a binary tree. We define the initial set $D_0$ as $\Sigma$, representing the base alphabet. The set $D_n$ is then recursively constructed, where $n$ denotes the tree depth of a regex in $D_n$. When constructing $D_n$, we apply unary operations to elements in $D_{n-1}$ and binary operations between elements from $D_{\leq n-1}$ and $D_{n-1}$. Binary operations are restricted to cases where the operand from $D_{\leq n-1}$ comes first in binary operations to control dataset growth. The construction process is described in Algorithm 1. Since the number of regexes grows exponentially with the depth, we limit LRD to regexes of depths up to 3. The dataset is partitioned into train, validation, and test sets in a ratio of 20:2:1. The test set size is relatively smaller to reduce computational costs during LLM inference. Table 3 provides dataset statistics.

Given a regex, all smaller regexes must be examined to check equivalence to determine the minimal tree length. Similar to the construction of $D_n$, we initialize $A_1 = \Sigma$ and recursively construct $A_n$, where $n$ represents the tree length of regexes in $A_n$. Unary operations are applied to $A_{n-1}$, while binary operations are applied to all possible pairs between elements from $A_i$ and $A_{n-i-1}$. The detailed process is described in Algorithm 4. This recursive definition ensures completeness, with a proof provided in Appendix D.1. Once $A_{\leq n}$ is constructed, regexes are partitioned into

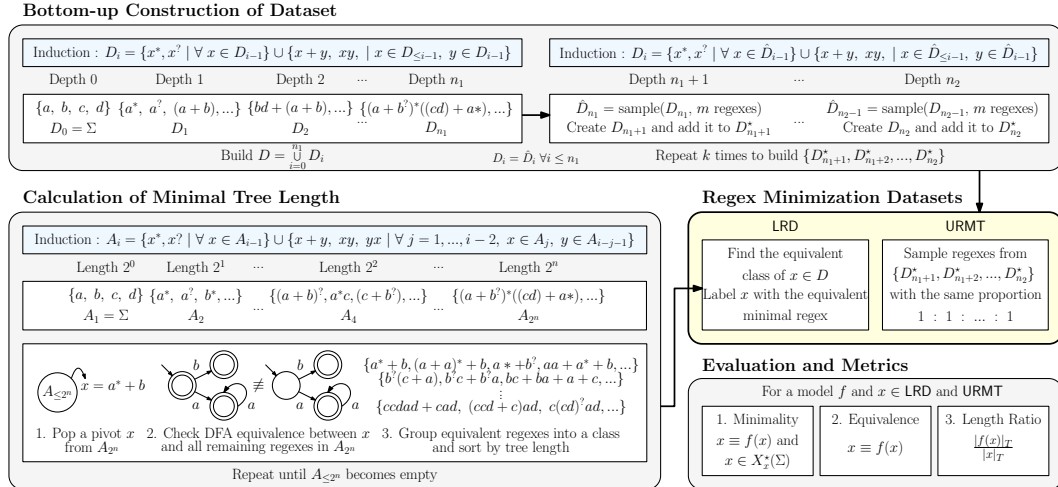

Figure 5: The overview of our dataset construction and evaluation. Our dataset consists of regex minimization corpus (LRD) and extended regex minimization benchmark (URMT), which are constructed using a bottom-up approach over tree depth. LRD is labeled using the minimal tree length calculated in Section B.1.

---

**Algorithm 1** Construction of LRD

**Require:** Maximum depth $n$, alphabet $\Sigma$
  Initialize $D_0 = \Sigma$
  **for** $i = 1 \ldots n$ **do**
    $D_i \leftarrow \emptyset$
    **for** $x \in D_{i-1}$ **do**
      $D_i \leftarrow D_i \cup \{\text{concat}(x, \text{`?'})\}$
      $D_i \leftarrow D_i \cup \{\text{concat}(x, \text{`*'})\}$
    **end for**
    **for** $x \in D_{\leq i-1}$ **do**
      **for** $y \in D_{i-1}$ **do**
        $D_i \leftarrow D_i \cup \{\text{concat}(x, \text{`+'}, y)\}$
        $D_i \leftarrow D_i \cup \{\text{concat}(x, y)\}$
      **end for**
    **end for**
  **end for**

---

equivalence classes. We select a regex from $A_{\leq n}$ as a pivot, and all equivalent regexes are grouped into the same class. The minimal tree length for each class is determined as the smallest tree length within the group. The proof of minimality and the equivalence class partitioning algorithm are in Appendix D.1.

This approach requires a quadratic number of comparisons, each taking exponential time, making it computationally infeasible. We reduce the computation by applying a heuristic based on string acceptance. This method generates sequences of strings that a regex may accept, allowing regexes to be partitioned based on acceptance and rejection. Once the regex set is sufficiently reduced, the remaining comparisons are performed to determine equivalence. This method also helps in determining whether a regex $x$ has an equivalent regex of length $n$ or less by narrowing down candidate equivalent regexes. Using this approach, LRD is labeled. Examples of our LRD are in Appendix C.2.

As mentioned earlier in this section, the process of calculating the equivalence of all regex pairs requires $O(n^2)$ equivalence comparisons. We utilize string acceptance in order to reduce the number of candidates within a group that can be equivalent to one another. The detailed algorithm is described in Algorithm 2. For a given string $s$, a regex that accepts $s$ and another regex that rejects

$s$ cannot be equivalent. Based on this observation, we use string acceptance as a query to partition the regex group. First, we categorize regexes based on the alphabet character set they contain. Then, we check the acceptance of a predefined string sequence. Using the acceptance information of the string sequence as a binary encoding, we further divide the regexes into potentially equivalent classes. Once the group size becomes sufficiently small, we perform the $O(n^2)$ equivalence comparison to construct the equivalent classes. Although this procedure does not reduce the time complexity, it reduces the practical running time of partitioning regex groups into equivalent classes. Also, when a new regex to minimize is given as a query, we can find the candidate equivalent class by checking string acceptance of the given regex on the predefined string sequence.

---

**Algorithm 2** An algorithm for building $M_n$

---

**Require:** an alphabet $\Sigma, A_1, A_2, \ldots, A_n$, an integer $m$ and the fixed sequence of strings S

   $A \leftarrow \bigcup_{i=1}^{n} A_i$
   $G \leftarrow map()$
   **for** $\sigma \in 2^{\Sigma}$ **do**
      $G[\sigma] \leftarrow \emptyset$
      label $G[\sigma]$ as 1
   **end for**
   **while** $|A| > 0$ **do**
      $x \leftarrow A.pop()$
      $\sigma_x \leftarrow$ the alphabet of $x$
      $G[\sigma_x] \leftarrow G[\sigma_x] \cup \{x\}$
   **end while**
   $H \leftarrow G.values()$
   **while** $|H| > 0$ **do**
      $X \leftarrow H.pop()$
      **if** $|X| > m$ **then**
         $s \leftarrow S[X.label]$
         $X_{success} \leftarrow \emptyset$
         label $X_{success}$ as X.label $\ll$ 1+1
         $X_{fail} \leftarrow \emptyset$
         label $X_{fail}$ as X.label $\ll$ 1
         **for** $y \in X$ **do**
            **if** $y$ accepts $s$ **then**
               $X_{success} \leftarrow X_{success} \cup \{y\}$
            **else**
               $X_{fail} \leftarrow X_{fail} \cup \{y\}$
            **end if**
         **end for**
         $H \leftarrow H \cup \{X_{success}, X_{fail}\}$
      **else**
         **while** $|X| > 0$ **do**
            $x \leftarrow X.pop()$
            **for** $y \in X$ **do**
               **if** $y \equiv x$ **then**
                  $X.remove(y)$
                  **if** $|y|_T < |x|_T$ **then**
                     $x \leftarrow y$
                  **end if**
               **end if**
            **end for**
            $M_n \leftarrow M_n \cup \{x\}$
         **end while**
      **end if**
   **end while**

---

**Algorithm 3** Construction of URMT

**Require:** Number of iterations $k$, sample size $m$, depth range $[n_1, n_2]$, alphabet $\Sigma$
  **for** $i = 1 \dots n_2$ **do**
    Initialize $D_i^\star \leftarrow \emptyset$
  **end for**
  **for** $t = 1 \dots k$ **do**
    Initialize $\hat{D}_0 = \Sigma$
    **for** $i = 1 \dots n_2$ **do**
      Initialize $D_i \leftarrow \emptyset$
      **for** $x \in \hat{D}_{i-1}$ **do**
        $D_i \leftarrow D_i \cup \{\text{concat}(x, \text{`?'})\}$
        $D_i \leftarrow D_i \cup \{\text{concat}(x, \text{`*'})\}$
      **end for**
      **for** $x \in \hat{D}_{\leq i-1}$ **do**
        **for** $y \in \hat{D}_{i-1}$ **do**
          $D_i \leftarrow D_i \cup \{\text{concat}(x, \text{`+'}, y)\}$
          $D_i \leftarrow D_i \cup \{\text{concat}(x, y)\}$
        **end for**
      **end for**
      $D_i^\star \leftarrow D_i^\star \cup D_i$
      $\hat{D}_i \leftarrow$ sample $m$ regexes from $D_i$
    **end for**
  **end for**
  **for** $i = n_1 \dots n_2$ **do**
    $D_i^{final} \leftarrow$ sample from $D_i^\star$
  **end for**

## B.2 URMT Construction

The dataset $D_{\leq 3}$ consists of short regexes, making it necessary to evaluate performance on longer, unseen examples to assess generalizability. We provide an unlabeled benchmark with regexes of depths 4 to 6 to address this. For depths 4 to 6, we construct the sets by sampling $m\ (= 1000)$ regexes from the previous depth before applying binary operations. However, this direct sampling may cause repeated patterns, which distorts the distribution of the benchmark. In order to mitigate this, we repeat the construction process $k\ (= 10)$ times, combine the results, and then sample regexes in a 1:1:1 ratio by depth. The construction process is described in Algorithm 3.

## B.3 REGEXPSPACE BENCHMARK CONSTRUCTION

We conduct preliminary experiments on the RegexMin task, as described in Appendix F. The results show that the overall performance of LLMs remain poor, indicating significant difficulty in solving this task. Even when prompted with simple instructions or Chain-of-Thought prompting, the performance has improved marginally. Moreover, experiments with Bart and T5 models trained from scratch, as well as fine-tuned LLMs, demonstrate strong performance on labeled datasets of lengths similar to those seen during training, but their performance degrades severely on URMT. In contrast, proprietary LLMs exhibited far superior reasoning abilities when tested on a small set of examples where other LLMs consistently failed. In particular, models explicitly categorized as reasoning models outperformed their non-reasoning counterparts by a substantial margin.

Based on these findings, we concluded that this task offers clear advantages for evaluating the reasoning performance of LLMs, and accordingly, we constructed the RegexPSPACE benchmark. This benchmark was derived from the test split of the labeled regex dataset containing 50,000 regexes, to which we applied a series of filtering criteria. The four criteria were as follows:

- The regex must not already be minimal

- The equivalence class obtained during construction must contain at least 10 regexes.

- Each regex must have at least 20 positive and 20 negative examples.

- If multiple regexes are equivalent under one-to-one alphabet mapping (iso morphism), only one is retained through random sampling.

These criteria were selected to avoid trivial regular expressions and overly simple tasks.

After applying them, we obtain 1,685 regexes from the original 50,000. In order to support the equivalence task, candidate regex pairs were required. For each of the 1,685 regexes, we label one equivalent regex and one non-equivalent regex to ensure a balanced dataset. Equivalent regexes are sampled from those generated during the minimization process. Non-equivalent regexes are constructed by generating permutations from the set of regexes with the same alphabet within the benchmark, ensuring that no regex is paired with itself, thereby guaranteeing a certain degree of hardness.

## C  DETAILS OF DATASET AND BENCHMARK

### C.1  DATASET STATISTICS

In this section, we present detailed statistics of the dataset we constructed. The number of regexes, their lengths, and the applicable tasks for each split are summarized in Table 3.

Table 3: Dataset statistics of LRD and URMT. LRD consists of train, validation, and test sets, while URMT consists of test set.

| Dataset | # of regexes | | | Depth | Length | Task |
|---------|-------|------------|------|-------|--------|------|
|         | Train | Validation | Test |       |        |      |
| RegexPSPACE | - | - | 1,685 | $\leq 3$ | $\leq 15$ | RegexEq, RegexMin |
| LRD | 1,100,000 | 116,752 | 50,000 | $\leq 3$ | $\leq 15$ | RegexEq, RegexMin |
| URMT | - | - | 50,000 | $\leq 6$ | $\leq 127$ | RegexMin |

### C.2  DATASET EXAMPLES

Table 4 presents examples from each of our constructed regex datasets. We report query regex examples for depths ranging from 0 to 6. As shown in the table, our regexes are fully parenthesized to explicitly represent depth, whereas the outputs are not. We do not provide the depth of the minimal regex, since tree depth can vary even for identical regexes depending on how parentheses are structured.

Table 4: Examples of our regex minimization dataset. We report the example regexes of depth 0 to 6 sampled from LRD and URMT. LRD contains the minimal equivalent of a regex and the minimal length, while URMT only contains regexes with their lengths.

| LRD, RegexPSPACE | | |
|---|---|---|
| Depth | Field | Content |
| 0 | Query Regex | $b$ |
| | Query Length | 1 |
| | Minimal Regex | $b$ |
| | Minimal Length | 1 |
| | Equivalent Regex | $-$ |
| 1 | Query Regex | $ab$ |
| | Query Length | 3 |
| | Minimal Regex | $ab$ |
| | Minimal Length | 3 |
| | Equivalent Regex | $-$ |
| 2 | Query Regex | $a^* + a + d$ |
| | Query Length | 6 |
| | Minimal Regex | $d + a^*$ |
| | Minimal Length | 4 |
| | Equivalent Regex | $d + (aa + (aa)^*)a^?$ |
| 3 | Query Regex | $((a + (b^*)) + ((a^?) + (b^?)))$ |
| | Query Length | 10 |
| | Minimal Regex | $a + b^*$ |
| | Minimal Length | 4 |
| | Equivalent Regex | $a + b + (bbb^?b^?)^*$ |
| URMT | | |
| Depth | Field | Content |
| 4 | Query Regex | $(d + c)c^? + ca + c + c + b^* + a^* + bd(b + a)$ |
| | Query Length | 28 |
| 5 | Query Regex | $(ac + a + d + (a + d)c)adaacdc$ $(dda^? + b + d + b^* + (a + b)c^* + b + c + a^?)$ |
| | Query Length | 55 |
| 6 | Query Regex | $(b^* + c + b + ba^* + c^?(b + b) + (a + c)b^?)$ $(c^*cb(a + c + cb) + a + c + c^* + bd^*)$ $+(c^* + a + a)(d + b)ccc^?(b + db)$ $+dd + c^* + d + aba^* + c + d + d$ |
| | Query Length | 98 |

# D PROOFS FOR CORRECTNESS OF DATASET

## D.1 PROOF FOR MINIMALITY

In this section, we discuss algorithms and proofs that were not covered in Section B. More specifically, our goal is to prove the minimality of the constructed dataset. The proof sketch is as follows: $A_n$ generated by Algorithm 4, and $M_n$ generated by Algorithm 5, contain all possible regexes of length $n$ or less. Moreover, the sequences $M_1, M_2, \ldots, M_n$ follows an inclusion relationship $M_1 \subset M_2 \subset \ldots \subset M_n$, where it contains all $M_i$ for $i \leq n$ as $n$ increases. Thus, any regex of length $n$ or less has an equivalent minimal regex included in some $A_i$ for $i \leq n$, and by the inclusion relationship, it must be contained in $M_n$. This proves that $M_n$ contains all minimal regexes of length $n$ or less. Furthermore, based on the construction method, we can show that non-minimal regexes are not included in $M_n$. This implies that $M_n$ represents the complete set of all minimal regexes of length $n$ or less.

**Lemma D.1.** *Given an integer $n$ and an alphabet $\Sigma$, the set $A_n$ constructed by algorithm 4 contains all possible regular expressions on $\Sigma$, when considering the equivalence relation as the same.*

---

**Algorithm 4** An algorithm for building $A_n$

---

**Require:** integer $n$, alphabet $\Sigma$
   Initialize $A_1 = \Sigma$
   **for** $i = 2 \dots n$ **do**
      $A_i \leftarrow \emptyset$
      **for** $x \in A_{i-1}$ **do**
         $A_i \leftarrow A_i \cup \{\text{concat}(x, \text{`?'})\}$
         $A_i \leftarrow A_i \cup \{\text{concat}(x, \text{`*'})\}$
      **end for**
      **for** j = $1 \dots \frac{i-1}{2}$ **do**
         **for** $x \in A_j$ **do**
            **for** $y \in A_{i-1-j}$ **do**
               **if** $x \equiv y$ **then**
                  $A_i \leftarrow A_i \cup \{\text{concat}(x, y)\}$
               **else**
                  $A_i \leftarrow A_i \cup \{\text{concat}(x, \text{`+'}, y)\}$
                  $A_i \leftarrow A_i \cup \{\text{concat}(x, y)\}$
                  $A_i \leftarrow A_i \cup \{\text{concat}(y, x)\}$
               **end if**
            **end for**
         **end for**
      **end for**
   **end for**

---

*Proof.* Assume that there exists an regular expression $x$ over $\Sigma$ such that $x \notin A_n$, and $|x|_T = n$. Each regular expression can be represented by a binary tree. Denote the binary tree notation of a regular expression $x$ be $T(x)$, and the left and right subtrees of $x$ be $x_{left}$ and $x_{right}$. Let $|x_{left}|_T$ be $j(\geq 1)$. Since $x \notin L_n$, at least one of $x_{left} \notin A_j$ or $x_{right} \notin A_{n-1-j}$ is true. By using the recursion, we can conclude that there exists a single character $c \in \Sigma$, which is included as a character in the regex $x$, but $c \notin A_1$. This contradicts the fact that $A_1 = \Sigma$. Thus, there is no regular expression $x \notin A_n$ with $|x|_T = n$. $\qquad\square$

---

**Algorithm 5** An algorithm for building $M_n$

---

**Require:** $A_1, A_2, \dots, A_n$
   $A \leftarrow \bigcup\limits_{i=1}^{n} A_i$
   $M_n \leftarrow List()$
   **while** $|A| > 0$ **do**
      $x \leftarrow A.\text{pop}()$
      **for** $y \in A$ **do**
         **if** $y \equiv x$ **then**
            $A.\text{remove}(y)$
            **if** $|y|_T < |x|_T$ **then**
               $x \leftarrow y$
            **end if**
         **end if**
      **end for**
      $M_n \leftarrow M_n \cup \{x\}$
   **end while**

---

**Lemma D.2.** *Given the sets $A_1, A_2, \dots, A_n$ constructed by algorithm 4 and an alphabet $\Sigma$, the sets $M_n$ constructed by algorithm 5 contains all possible regular expressions on $\Sigma$, when considering the equivalence relation as the same.*

*Proof.* Assume that there is a regular expression $x \notin M_n$ with $|x|_T \leq n$. By lemma D.1, $x \in \bigcup_{j=1}^{n} A_j$, and it must be considered by the loop in line 3 of algorithm 5. Then, $x$ should be in $M_n$, since the representative of each equivalent class in $\bigcup_{j=1}^{n} A_j$ is added to $M_n$. This contradicts the assumption that $x \notin M_n$. Thus, $M_n$ contains all possible regular expressions on $\Sigma$. $\qquad\square$

**Lemma D.3.** *Given the sets $A_1, A_2, \ldots, A_i, \ldots, A_n$ constructed by algorithm 4 and an alphabet $\Sigma$, the sets $M_i$ and $M_n$ are constructed by algorithm 5. Then, $M_i \subset M_n \ \forall \ i = 1, \ldots, n$, when considering the equivalence relation with the same tree length as the same.*

*Proof.* Assume that there is a regular expression $x \in M_i$ with $|x|_T = i \leq n$, but $x \notin M_n$, when considering the equivalence relation with the same tree length as the same. By the lemma D.2, there exists an equivalent regular expression $x' \in M_n$ with $|x'|_T = j \neq i$. Note that $i, j \leq n$.

**Case1:** $i < j$ By the lemma D.1, $x \in \bigcup_{k=1}^{i} L_k \subset \bigcup_{k=1}^{n} L_k$. When we compare the tree lengths of regular expressions during the construction of $M_n$ by algorithm 5, $x'$ must be discarded and replaced by $x$. Thus, $j$ cannot be larger than $i$.

**Case2:** $j < i$ By the lemma D.2, $x' \in \bigcup_{k=1}^{i} A_j \subset \bigcup_{k=1}^{i} A_k$. When we compare the tree lengths of regular expressions during the construction of $M_i$ by algorithm 5, $x$ must be discarded and replaced by $x'$. Thus, $i$ cannot be larger than $j$

Since $i \geq j$ and $i \leq j$, $i$ and $j$ must be the same, which contradicts the assumption. Thus, $M_i \subset M_n$ $\forall i = 1, \ldots, n$. $\qquad\square$

**Corollary D.4.** *Given the sets $A_1, A_2, \ldots, A_n$ constructed by algorithm 4 and an alphabet $\Sigma$, the set $M_n$ constructed by algorithm 5 contains all possible minimal regular expressions on $\Sigma$ with $|x|_T \leq n$, when considering the equivalence relation with the same tree length as the same.*

*Proof.* Assume that there is a minimal regular expression $x$ with $|x|_T = i \leq n$ such that $x \notin M_n$. By the lemma D.2, $x \in M_i$. Then, by the lemma D.3 $x \in M_n$. This contradicts the assumption. Thus, there is no such $x$. $\qquad\square$

**Corollary D.5.** *Given the sets $A_1, A_2, \ldots, A_n$ constructed by algorithm 4 and an alphabet $\Sigma$, the set $M_n$ constructed by algorithm 5 contains only minimal regular expressions on $\Sigma$ with $|x|_T \leq n$, when considering the equivalence relation with the same tree length as the same.*

*Proof.* Assume that there is a non-minimal regular expression $x \in M_n$. There is an minimal regular expression $x'$ of $x$ with $|x'|_T = i \leq n$. Then, $x'$ must be in $M_n$ by corollary D.4. It contradicts the assumption because both $x$ and $x'$ cannot be in $M_n$ by the algorithm 5. Thus, there is no such $x$. $\qquad\square$

**Theorem D.6.** *Given the sets $A_1, A_2, \ldots, A_n$ constructed by algorithm 4, an alphabet $\Sigma$, and the set $M_n$ constructed by algorithm 5, the follwing statement holds, when considering the equivalence relation with the same tree length as the same.*

$$x \in M_n \ \ if \ \ and \ \ only \ \ if \ \ x \in X_x^{\star}(\Sigma) \ \ where \ \ |x|_T \leq n$$

*Proof.* It is proved by combining Corollary D.4 and D.5. $\qquad\square$

# E    COMPUTATIONAL COST OF DATASET CONSTRUCTION

Our dataset size is determined by the need to exhaustively compare each regex with all shorter expressions to verify minimality. As the depth or length of regexes increases, the number of candidate expressions grows double exponentially due to our inducting construction. We report the computational cost over different sizes of alphabet and different depth. Each following Tables 5 and 6

contains the number of regexes in the dataset and the number of regexes we need to investigate to find the minimal regexes. Note that the number of regexes grows in double exponential which makes it almost impossible to increase depth more than 3. We construct the dataset based on the induction of the depth which is restricted to 3 allowing the regular expressions with the length at most 15. Although the dataset may appear small, we exhaustively compare a vast number of regexes to identify the minimal ones, a process that is extremely labor intensive.

Table 5: The number of regexes contained in the dataset constructed by the proposed procedure depending on the size of alphabet and depth

| Size of Alphabet ($|\Sigma|$) | Depth | | | | | |
|---|---|---|---|---|---|---|
| | 0 | 1 | 2 | 3 | 4 | 5 |
| 2 | 2 | 10 | 170 | 3.35E+04 | 1.13E+09 | 1.29E+18 |
| 3 | 3 | 18 | 486 | 2.58E+05 | 6.69E+10 | 4.47E+21 |
| 4 | 4 | 28 | 1092 | **1.27E+06** | 1.60E+12 | 2.57E+24 |
| 5 | 5 | 40 | 2120 | 4.69E+06 | 2.20E+13 | 4.85E+26 |
| 6 | 6 | 54 | 3726 | 1.43E+07 | 2.06E+14 | 4.23E+28 |
| 7 | 7 | 70 | 6090 | 3.80E+07 | 1.45E+15 | 2.10E+30 |
| 8 | 8 | 88 | 9416 | 9.05E+07 | 8.19E+15 | 6.71E+31 |
| 9 | 9 | 108 | 13932 | 1.97E+08 | 3.90E+16 | 1.52E+33 |

Table 6: The number of regexes to be investigated to find minimal by the proposed procedure depending on the size of alphabet and depth

| Size of Alphabet ($|\Sigma|$) | Depth | | | | | |
|---|---|---|---|---|---|---|
| | 0 | 1 | 2 | 3 | 4 | 5 |
| 2 | 2 | 20 | 6176 | 2.16E+09 | 8.46E+20 | 3.85E+44 |
| 3 | 3 | 39 | 20118 | 2.10E+10 | 7.63E+22 | 3.00E+48 |
| 4 | 4 | 64 | 48640 | **1.15E+11** | 2.21E+24 | 2.46E+51 |
| 5 | 5 | 95 | 98870 | 4.52E+11 | 3.31E+25 | 5.39E+53 |
| 6 | 6 | 132 | 179232 | 1.42E+12 | 3.20E+26 | 4.95E+55 |
| 7 | 7 | 175 | 299446 | 3.80E+12 | 2.26E+27 | 2.44E+57 |
| 8 | 8 | 224 | 470528 | 9.05E+12 | 1.26E+28 | 7.52E+58 |
| 9 | 9 | 279 | 704790 | 1.96E+13 | 5.86E+28 | 1.61E+60 |

# F    PRELIMINARY EXPERIMENTS ON REGEXMIN

As a form of preliminary experiments, we evaluate three settings on the minimization task: fine-tuned LLMs, small models trained from scratch, and pretrained LLMs.

## F.1    PRELIMINARY EXPERIMENTS

We report the experimental results on LRD in Table 7 and on URMT in Table 8. Results reveal that pretrained LLMs exhibit poor performance, while fine-tuned and scratch-trained models perform reasonably well on regexes of lengths similar to those seen during training but fail to generalize to longer inputs. Notably, although performance improves with larger LLMs, they still fail to minimize effectively beyond a few simple heuristics. In constrast, a small case study on proprietary LLMs demonstrates that some models can solve minimization tasks without additional training, and proprietary reasoning models in particular perform well on longer regexes. This suggests that the ability to handle longer contexts is closely tied to reasoning capacity and there is a quite large margin between open-source and proprietary LLMs to improve.

Table 7: Main results on LRD. We evaluate two LMs trained on LRD from scratch and five LLMs from Llama and Qwen family with zero-shot and five-shot prompting. We report the minimality, equivalence, and length ratio as metrics. The best performance are bolded in each trained LMs and pre-trained LLMs.

| Approach | Model | Size | Shot | Min. (↑) | Equiv. (↑) | Ratio (↓) |
|---|---|---|---|---|---|---|
| Training | BART | 139M | - | **85.89**±0.07 | **87.12**±0.14 | **79.66**±0.02 |
| | T5 | 223M | - | 85.56±1.13 | 86.71±1.16 | 79.72±0.15 |
| Finetuning | Llama3.1 | 8B | - | **99.99**±0.00 | **100.00**±0.00 | **77.94**±0.00 |
| | Qwen2.5 | 7B | - | 98.93±0.19 | 98.93±0.19 | 78.15±0.04 |
| Prompting | Llama3.1 | 8B | Zero | 1.72 | 5.88 | 99.39 |
| | | | Five | 3.95 | 10.82 | 98.54 |
| | Llama3.3 | 70B | Zero | 13.45 | 24.27 | 97.33 |
| | | | Five | 21.15 | 39.82 | 94.30 |
| | Qwen2.5 | 7B | Zero | 7.63 | 16.51 | 98.91 |
| | | | Five | 6.75 | 15.66 | 98.17 |
| | Qwen2.5-coder | 7B | Zero | 10.48 | 18.43 | 97.85 |
| | | | Five | 10.67 | 24.69 | 97.26 |
| | Qwen2.5 | 72B | Zero | 20.31 | **54.52** | 95.37 |
| | | | Five | **23.06** | 54.33 | **93.09** |
| CoT | Llama3.3 | 70B | Zero | 23.25 | 37.79 | 93.31 |
| | | | Five | 29.07 | 41.82 | 92.48 |
| | Qwen2.5 | 72B | Zero | 19.09 | 44.41 | 91.82 |
| | | | Five | **31.13** | **46.87** | **90.70** |

Table 8: Main results on URMT. We evaluate two LMs trained on LRD from scratch and five LLMs from Llama and Qwen family with zero-shot and five-shot prompting. We report the equivalence, and length ratio as metrics. The best performance are bolded in each trained LMs and pre-trained LLMs.

| Approach | Model | Size | Shot | Equiv. (↑) | Ratio (↓) |
|---|---|---|---|---|---|
| Training | BART | 139M | - | **0.08**±0.01 | **99.93**±0.01 |
| | T5 | 223M | - | 0.05±0.01 | 99.95±0.01 |
| Finetuning | Llama3.1 | 8B | - | 0.86±0.09 | 99.44±0.04 |
| | Qwen2.5 | 7B | - | **1.23**±0.07 | **99.34**±0.01 |
| Prompting | Llama3.1 | 8B | Zero | 1.51 | 99.96 |
| | | | Five | 0.26 | 99.99 |
| | Llama3.3 | 70B | Zero | 7.16 | 99.60 |
| | | | Five | 2.57 | 99.73 |
| | Qwen2.5 | 7B | Zero | 1.79 | 99.90 |
| | | | Five | 0.66 | 99.95 |
| | Qwen2.5-coder | 7B | Zero | 3.28 | 99.85 |
| | | | Five | 1.36 | 99.95 |
| | Qwen2.5 | 72B | Zero | **13.99** | **99.28** |
| | | | Five | 6.44 | 99.50 |

## F.2 EXPERIMENTAL CASE STUDY ON PROPRIETARY MODELS IN REGEXMIN

This section examines how well state-of-the-art LLMs perform in regex minimization based on a small test set. Using the five-shot prompt, we evaluate GPT-4o, one of the most popular LLMs, GPT-o1, known for its reasoning capabilities, and DeepSeek-R1, which has recently gained significant attention as a competitive open LLM. The evaluation set consists of five regexes from the LRD test set, selected to highlight common failure patterns of LLMs observed, along with two regexes from each of depths 4, 5, and 6 in URMT, resulting in a total of eleven test cases.

The experimental results are in Table 9 and 10. GPT-4o successfully produces equivalent regexes in most cases from LRD but reveal clear limitations in regex minimization. As the regex length increases, it fails to generate equivalent outputs. DeepSeek-R1 successfully minimizes all regexes

in LRD and handles equivalent regex generation up to depth 4. However, for regexes of depth 5 or higher, it struggles to produce equivalent outputs. GPT-o1 successfully minimizes all regexes in LRD and generates equivalent regexes for all URMT examples.

Despite being considered models with strong reasoning capabilities, both GPT-4o and DeepSeek-R1 exhibit clear limitations in reasoning when applied to regex minimization. This raises the question of whether existing LLMs, even those optimized for reasoning, may not be truly reasoning in the context of structured, algorithmic tasks. At the same time, these findings demonstrate that The consistent performance degradation observed from long regexes in URMT underscores the complexity of regex minimization and further demonstrates the difficulty of our dataset as a valuable benchmark. The results highlight the need for explicit mechanisms that enhance structured reasoning, as LLMs struggle with systematic transformations required for regex minimization.

# G    EVALUATION DETAILS

## G.1    DETAILS OF LLMS USED FOR EVALUATION

Our evaluation utilizes a diverse set of LLMs to assess performance across various architectures and training methodologies. The models are distinguished into non-reasoning models and reasoning models. Non-reasoning models include foundational models known for their strong performance in various downstream tasks such as question and answering, common sense reasoning, mathematical reasoning, and code generation. without specific fine-tuning. Reasoning models include LLMs that are either explicitly designed for complex reasoning or have demonstrated exceptional performance on reasoning benchmarks.

### G.1.1    NON-REASONING MODELS

**Llama-3**   Llama-3 is a family of LLMs developed and released by Meta AI (et al., 2024a). As the successor of Llama-2, Llama-3 represents one of the state-of-the-art open-source LLMs, trained on a significantly larger and more diverse dataset. We utilize Llama-3.1 of 8B size in our experiments on evaluating RegexEq and RegexMin.

**Qwen**   Qwen is an open-source and multilingual foundation model series developed by Alibaba Cloud (et al., 2023). Qwen2.5 technical report (et al., 2024b) details post-training with over one million supervised examples and multi-stage reinforcement learning (RL), along with the model sizes from 0.5B to 72B, and strong performance across language, math, and coding. The family also provides 1M-token long-context vairants (et al., 2025a). In our main evaluation, we use Qwen2.5 with 7B and 14B sizes, and we also conduct experiments on Qwen3-A3B with 30B parameters for additional evaluations in Appendix H.2.

**Qwen-Coder**   Qwen2.5-Coder is a code-specialized branch of Qwen2.5 trained on over 5.5 trillion additional code-centric tokens and released in six sizes from 0.5B to 32B (Hui et al., 2024). It reports state-of-the-art results among open-source models on multiple code-generation and repair benchmarks, while maintaining general language and math skills. We use Qwen2.5-Coder with 7B and 14B sizes as our non-reasoning baselines for the main evaluation and also conduct experiments over Qwen3-Coder-A3B of 30B size for an analysis in Appendix H.2.

**Phi-4**   Phi-4 is a 14B-parameter, decoder-only open-source model developed by Microsoft. It is designed to provide strong quality at a small scale via carefully curated and synthetic training data; it is distributed with an emphasis on safety-aligned post-training(Abdin et al., 2024). We use this model as an LLM baseline.

**EXAONE**   EXAONE, developed by LG AI Research, is a multilingual model family that supports English, Korean, and Spanish. The latest version, EXAONE-4.0 Bae et al. (2025) enhances its capabilities for advanced logic and agentic tool usage. We use EXAONE-4.0 with 32B parameters for an empirical study in Appendix H.2.

| LRD | |
|---|---|
| Field | Content |
| Query Regex | $(((a + b) + (b + c)) + ((b + d) + (c + c)))$ |
| Minimal Regex | $(a + b + c + d)$ |
| GPT-4o Output | $(a + b + c + d)$ |
| Evaluation | Minimal, Length Ratio: 0.4667 |
| DS-R1 Output | $(a + b + c + d)$ |
| Evaluation | Minimal, Length Ratio: 0.4667 |
| GPT-o1 Output | $(a + b + c + d)$ |
| Evaluation | Minimal, Length Ratio: 0.4667 |
| Query Regex | $((a + (a + b)) + ((a + b)(b + c)))$ |
| Minimal Regex | $((a + b)(b + c)^?)$ |
| GPT-4o Output | $(a + (a + b)(b + c))$ |
| Evaluation | Inequivalent |
| DS-R1 Output | $(a + b)(b + c)^?$ |
| Evaluation | Minimal, Length Ratio: 0.6154 |
| GPT-o1 Output | $(a + b)(b + c)^?$ |
| Evaluation | Minimal, Length Ratio: 0.6154 |
| Query Regex | $(((a^?)(b^*)) + ((b^*)(b^?)))$ |
| Minimal Regex | $(a^?b^*)$ |
| GPT-4o Output | $(a^?b^* + b^*b^?)$ |
| Evaluation | Equivalent, Length Ratio: 1.0000 |
| DS-R1 Output | $(a^?b^*)$ |
| Evaluation | Minimal, Length Ratio: 0.4545 |
| GPT-o1 Output | $a^?b^*$ |
| Evaluation | Minimal, Length Ratio: 0.4545 |
| Query Regex | $((d + (a^?)) + ((b^*) + (b^?)))$ |
| Minimal Regex | $(a + d + b^*)$ |
| GPT-4o Output | $(d + a^? + b^*)$ |
| Evaluation | Equivalent, Length Ratio: 0.7000 |
| DS-R1 Output | $(a + d + b^*)$ |
| Evaluation | Minimal, Length Ratio: 0.6000 |
| GPT-o1 Output | $(a + d + b^*)$ |
| Evaluation | Minimal, Length Ratio: 0.6000 |
| Query Regex | $(((b^*)(bb))((d^*)(d^*)))$ |
| Minimal Regex | $(bbb^*d^*)$ |
| GPT-4o Output | $(b^*bb)(d^*)$ |
| Evaluation | Minimal, Length Ratio: 0.7500 |
| DS-R1 Output | $((b^*bb)d^*)$ |
| Evaluation | Minimal, Length Ratio: 0.7500 |
| GPT-o1 Output | $((b^*bb)d^*)$ |
| Evaluation | Minimal, Length Ratio: 0.7500 |

Table 9: Case study of state of the art LLMs on LRD. We report case study using GPT-4o, GPT-o1, and DeepSeek-R1 on LRD test set. We report the type of failure and length ratio for each example. GPT-4o, the only model without reasoning ability, shows the lowest performance while others successfully minimize all the examples.

### G.1.2 REASONING MODELS

**Phi-4 Reasoning** Phi-4-reasoning is a 14B open-weight reasoning variant of Phi-4. It is fine-tuned on CoT traces and further aligned via RL, with a 32K context window (Abdin et al., 2025). The model reports competitive results on complex reasoning benchmarks such as AIME'24/25, GPQA-Diamond (Rein et al., 2023), LiveCodeBench (Jain et al., 2025), and MMLU-Pro (Hendrycks et al., 2021).

| URMT | |
|---|---|
| Query Regex | $((((b+a)(ca))((b^*)+c))+(((da)^*)((da)+d)))$ |
| GPT-4o Output | $((a+b)ca(b^*+c)+(da)^*d^?)$ |
| Evaluation | Inequivalent |
| DS-R1 Output | $((b+a)ca(b^*+c)+d(ad)^*a^?)$ |
| Evaluation | Equivalent, Length Ratio: 0.9565 |
| GPT-o1 Output | $((b+a)ca(b^*+c)+(da)^*da^?)$ |
| Evaluation | Equivalent, Length Ratio: 0.9565 |
| Query Regex | $((((cd)(a+b))+((c^?)(ab)))+(((bd)^*)((b+b)(bc))))$ |
| GPT-4o Output | $(cd(a+b)+c^?ab+bd^*(b+b)bc)$ |
| Evaluation | Inequivalent |
| DS-R1 Output | $(cd(a+b)+c^?ab+b(db)^*bc)$ |
| Evaluation | Equivalent, Length Ratio: 0.9259 |
| GPT-o1 Output | $(cd(a+b)+c^?ab+(bd)^*bbc)$ |
| Evaluation | Equivalent, Length Ratio: 0.9259 |
| Query Regex | $(((((a+a)(bd))+((cc)^?))(((ad)(b+b))((ad)(c^*))))(((((cc)(a+a))+(d^*))$ $+(((c+a)(bc))((aa)(bd))))))$ |
| GPT-4o Output | $(a+b)bd+(cc)^?(ad(b+b)adc^*+(cc(a+a)+d^*+(c+a)bcaabd))$ |
| Evaluation | Inequivalent |
| DS-R1 Output | $((abd+cc)^?adbadc^*(cca+d^*+(c+a)bcaabd))$ |
| Evaluation | Equivalent, Length Ratio: 0.8889 |
| GPT-o1 Output | $((abd+(cc)^?)adbadc^*(cca+d^*+(c+a)bcaabd))$ |
| Evaluation | Equivalent, Length Ratio: 0.8889 |
| Query Regex | $(((((c+c)^*)+((a^*)(b^*)))+(((dd)(b^?))(d+d)))+((((d+d)+(ad))+(d+(b+b)))$ $+((b+(dd))((d+d)+(d+d)))))$ |
| GPT-4o Output | $((c+c)^*+a^*b^*+dd(b^?)(d+d)+b+dd(d+d))$ |
| Evaluation | Inequivalent |
| DS-R1 Output | $(c^*+ab+d(a^?+b+db^?d))$ |
| Evaluation | Inequivalent |
| GPT-o1 Output | $(c^*+a^*b^*+ddb^?d+a^?d+bd)$ |
| Evaluation | Equivalent, Length Ratio: 0.5306 |
| Query Regex | $((((((d+a)(b^*))^*)+(((b+a)+(a+a))((b+b)+(bb))))(((a(ba))+((ba)(ba)))$ $+(((c^?)(c+c))+((d+c)+(da)))))(((((d+c)^*)+((d+b)+(d^*)))+(((b+a)(c^*))$ $+((a^?)(c^?))))+((((dc)+(bb))((d+b)(ac)))+(((db)(d^*))((bc)^*)))))$ |
| GPT-4o Output | $((d+a)(b^*)^*)+((a+b)(b+bb)+(ba)^*(c^?c+d+a))(c^*+d+b+d^*)(dc+bb)$ $(d+b)ac+(dbd^*)(bc)^*$ |
| Evaluation | Inequivalent |
| DS-R1 Output | $((((a+d)b^*)^*+(a+b)bb^?)((a+ba)ba+cc^?+da^?))+(d+c)^*+b$ $+(a+b)^?c^*+(dc+bb)(d+b)ac+dbd^*(bc)^*$ |
| Evaluation | Inequivalent |
| GPT-o1 Output | $((((((a+d)b^*)^*+(a+b)bb^?)(b^?aba+cc^?+da^?))((c+d)^*+(b+d^*)$ $+(a+b)c^*+a^?c^?+((bb+dc)(b+d)ac)+(dbd^*)(bc)^*))$ |
| Evaluation | Equivalent, Length Ratio: 0.8190 |
| Query Regex | $((((((c+d)d)+((c^*)+b))(((c^*)+(d^*))((a+d)(c+c))))(((((c+c)^?)$ $+((bd)(c+d)))+(((b^*)a)+((bb)(c+c)))))+(((((dd)+(b^*))$ $+((c+d)(c+d)))^?)(((((c+d)(bc))((dd)(ad)))+(((dd)(a+b))((c+d)(a^?)))))))$ |
| GPT-4o Output | $((c+d)d+c^*+b)((c^*+d^*)(a+d)cc)+(c+c)^?+bd(c+d)+(b^*a+bbcc)$ $+((dd+b^*+(c+d)(c+d))^?((c+d)bcddad+dda+b(c+d)a^?))$ |
| Evaluation | Inequivalent |
| DS-R1 Output | $((c^*+b+d(c+d))(c^*+d^*)(a+d)c(c^?+bd(c+d)+ba^*+bbc)$ $+(b^*+(c+d)(c+d))^?((c+d)bcddad+dd(a+b)(c+d)a^?))$ |
| Evaluation | Inequivalent |
| GPT-o1 Output | $((((((c+d)d+(c^*+b))((c^*+d^*)(a+d)c))((c^?+bd(c+d))+((b^*)a+bbc)))$ $+((((dd+b^*)+(c+d)(c+d)))^?(((c+d)bcddad)+((dd(a+b))((c+d)a^?)))))$ |
| Evaluation | Equivalent, Length Ratio: 0.9381 |

Table 10: Case study of state of the art LLMs on URMT. We report case study using GPT-4o, GPT-o1, and DeepSeek-R1 on URMT. We report the type of failure and length ratio for each example. GPT-4o show the lowest performance while GPT-o1 surpasses all other models.

**DeepSeek-R1** DeepSeek-R1 introduces an RL-first pipeline for reasoning: R1-Zero, trained with pure RL on a base model and R1, which employed RL with a small cold-start supervised fine-tuning (SFT) stage, followed by distillation into smaller dense models ranging from 1.5B to 70B.

DeepSeek-R1 reports strong pass@1 on AIME'24 and MATH-500, competitive GPQA-Diamond and high Codeforces ELO. The technical report (DeepSeek-AI, 2025) also provides detailed comparisons to OpenAI o1 and o1-mini.

**gpt-oss**  GPT-oss is OpenAI's open-source, reasoning-oriented mixture-of-experts (MoE) family with adjustable reasoning levels with 20B and 120B parameters. Likewise to the above two reasoning models, the official model card (et al., 2025b) includes comprehensive evaluation on AIME'24/25, GPQA-D, MMLU, SWE-Bench, Codeforces, and incldues safety assessments.

### G.2  DETAILS OF METRICS

#### G.2.1  MINIMIZATION TASK

We use three evaluation metrics for minimization tasks.

**Minimality**  The most intuitive metric is minimality, which represents the percentage of cases where the model successfully generates an equivalent minimal regex.

$$(\text{Minimality}) = \frac{\left|\{r \in D | f(r) \in X_r^\star(\Sigma)\}\right|}{|D|}$$

**Equivalence**  The second metric is equivalence, which measures the proportion of outputs that are equivalent to the input regex. Since many generated responses are either non-equivalent or syntactically invalid, we report the equivalence metric separately for a more detailed analysis. The minimality metric is stricter than the equivalence metric, as it considers only minimal and equivalent regexes as success.

$$(\text{Equivalence}) = \frac{\left|\{r \in D | f(r) \in X_r(\Sigma)\}\right|}{|D|}$$

**Length Ratio**  Additionally, we use Length Ratio, a metric adopted from previous work (Kahrs & Runciman, 2022) for regex simplification. The metric is based on the geometric mean of the length ratio between the original regex and minimized regex. When comparing lengths to calculate the length ratio, cases where the minimized regex is not equivalent to the original input or becomes longer are considered as not being reduced. In such cases, the ratio is set to 1, and the geometric mean is computed. Since URMT does not contain ground-truth minimized regexes, minimality cannot be measured. Instead, we report equivalence and length ratio for URMT.

$$(\text{Length Ratio}) = \big( \prod_{x \in D} \frac{|f(x)|_T}{|x|_T} \big)^{\frac{1}{|D|}}$$

#### G.2.2  EQUIVALENCE TASK

The evaluation of model performance on binary decision is grounded in the confusion matrix, which records the counts of true positives (TP), true negatives (TN), false positives (FP), and false negatives (FN). From this basis, the following standard metrics are defined.

**Accuracy**  Accuracy measures the overall proportion of correctly classified instances. While this provides an intuitive summary of correctness, it can be misleading in the presence of class imbalance or in binary classification settings, since high accuracy may be achieved simply by favoring the majority class.

$$(\text{Accuracy}) = \frac{TP + TN}{TP + TN + FP + FN}.$$

**Precision**   Precision captures the reliability of positive predictions. It quantifies the fraction of predicted positives that are truly positive, thus reflecting how well the model avoids false alarms.

$$(\text{Precision}) = \frac{TP}{TP + FP}.$$

**Recall**   Recall evaluates the ability of the model to identify all relevant positives. This metric reflects the model's sensitivity to true instances, a property of central importance in contexts where false negatives are especially costly.

$$(\text{Recall}) = \frac{TP}{TP + FN}.$$

**F1-score**   Precision and recall inherently exhibit a trade-off relationship. Increasing precision typically requires stricter decision boundaries, which reduces false positives but risks lowering recall by missing relevant instances. Conversely, relaxing these boundaries can improve recall by capturing more true positives, but at the expense of precision due to an increase in false positives. This tension highlights the importance of balancing the two metrics depending on the target application, as emphasizing one over the other can yield significantly different evaluation outcomes. The F1-score provides a single measure that balances precision and recall by taking their harmonic mean. This particularly valuable when neither precision nor recall alone fully characterizes performance, and when the dataset is imbalaced.

$$(\text{F1-score}) = 2 \times \frac{Precision \times Recall}{Precision + Recall}.$$

### G.3   Experimental Settings and Hyperparameters

In this section, we describe the specific libraries and experimental settings used in our work. From a hardware and system perspective, experiments are conducted with AMD Ryzen Threadripper 3960X CPUs, NVIDIA A6000 GPUs, and Rocky Linux 9.6 OS. The libraries and versions employed include Python 3.10, FAdo 2.2.0, CUDA 13.0, Torch 2.8.0, Huggingface Transformers 4.55.4, BitsAndBytes 0.47.0, Accelerate 1.2.1, Scikit-learn 1.7.2 and Unsloth 2025.3.19.

In order to handle large-scale LLM evaluatations, we employ NF4 quantization from BitsAndBytes and used greedy decoding. Ensuring fairness under the limitations of memory and computational resources, we constrain both the maximum number of thinking tokens and the maximum nubmer of answer tokens. For non-reasoning models, the maximum number of answer tokens is set to 1,024. For reasoning models, up to 4,096 thinking tokens are allowed, followed by 1,024 answer tokens. If the reasoning process did not naturally terminate, we insert a special token indicating the end of thinking before generating the 1,024 answer tokens.

For answer parsing, we instruct models to output their answers within specific tag, \boxed{}, via prompting. However, some models fail to adhere to the required format. Consequently, we apply several heuristic rules for parsing, which are summarized in the algorithm below.

### G.4   Prompts Used for Model Evaluation

We employ handcrafted prompts for LLM evaluation. In preliminary experiments on the labeled dataset, we observe that LLMs often produce practical regular expressions rather than formal ones. Practical regex differs from formal regex in terms of syntax and operators, particularly by allowing non-regular expressions (e.g., capture groups) and shorhand notations (e.g., [0-9] for a set of characters, Kleene plus, or exponent notation for character repetition). Such convenience-oriented operators are generally disallowed in the formal definition of regular expressions and may even violate regularity. Consequently, these expressions are not supported by the libraries we used or by prior works on regex simplification that we reference.

Nevertheless, LLMs often generate outputs in this practical form because they are frequently pretrained on practical regexes found in code. Therefore, we designed prompts that explicitly discour-

**Algorithm 6** An algorithm for parsing an LLM Output

---

**Require:** Model response string $s$

  **if** $s$ does not contain "boxed" **then**

    **for** line $l$ in $s$ (reversed) **do**

      **if** $l$ contains keyword (`answer:`, `answer**`) **then**

        **return** tokens after keyword

      **end if**

    **end for**

    **for** line $l$ in $s$ (reversed) **do**

      **if** $l$ contains indicative phrase (e.g., "minimal regex") **then**

        **return** substring after "is" or ":"

      **end if**

    **end for**

    **return** last line of $s$

  **else**

    Extract substring following "`boxed{`"

    Initialize stack with "{"

    Collect characters until delimiters are balanced

    $a \leftarrow$ extracted substring

  **end if**

  **if** $a$ contains "text" **then**

    Repeat stack-based extraction for "`text{`"

  **end if**

  Normalize $a$ by removing white spaces, ^, braces

  **if** $a$ contains '|' **then**

    $p \leftarrow$ True

    replace '|' with '+'

  **else**

    $p \leftarrow$ False

  **end if**

  **return** $a, p$

---

age the use of practical regex. Each prompt includes the formal definition of regular expressions, the definition of the specific task, and an explicit requirement that models should not produce practical regex. For ease of answer parsing, we also required models to enclose their final answer within a \boxed{} tag.

Depending on the task and few-shot setting, we prepared four types of prompt as provided in Figures 6, 7, 8, and 9. In the 5-shot setting, additional input-output examples are included. These few-shot examples were drawn from the training split using the same construction method as for the benchmark, and five fixed examples are consistently used.

---

**RegexMin Zero-shot**

---

You are an expert in formal regular expressions, commonly referred to as regex.

The formal regex must follow these rules:
- Allowed operations: concatenation, union (`+`), Kleene star (`*`), and option (`?`).
- Concatenation is implicit (no symbol is written).
- Parentheses `()` specify precedence.
- Do not use practical regex notations such as `|` for union or `+` for repetition.

Your task is to minimize a given formal regex over the fixed alphabet {`a`, `b`, `c`, `d`}. The minimized regex must be functionally equivalent to the input and have the smallest total number of symbols, where both characters (`a`, `b`, `c`, `d`) and operations (`+`, `*`, `?`), and concatenations are counted, but parentheses are not.

Enclose your final answer within `\\boxed{}`.

Minimize the following regex:

Input Regex: **$regex**
Output:

Figure 6: The zero-shot prompt used for RegexMin. It provides the formal definition of regular expressions, the task description, instructions prohibiting the use of practical regex, and formatting guidelines.

---

**RegexEq Zero-shot**

---

You are an expert in formal regular expressions, commonly referred to as regex.

The formal regex must follow these rules:
- Allowed operations: concatenation, union (`+`), Kleene star (`*`), and option (`?`).
- Concatenation is implicit (no symbol is written).
- Parentheses `()` specify precedence.
- Do not use practical regex notations such as `|` for union or `+` for repetition.

Your task is to determine whether two given formal regexes over the fixed alphabet {`a`, `b`, `c`, `d`} are equivalent. You must output either True or False.

Enclose your final answer within `\\boxed{}`.

Determine the equivalence of the following regexes:

Input Regex 1: **$regex1**
Input Regex 2: **$regex2**
Output:

Figure 7: The zero-shot prompt used for RegexEq. It provides the formal definition of regular expressions, the task description, instructions prohibiting the use of practical regex, and formatting guidelines.

---

**RegexMin Five-shot**

You are an expert in formal regular expressions, commonly referred to as regex.

The formal regex must follow these rules:
- Allowed operations: concatenation, union (`+`), Kleene star (`*`), and option (`?`).
- Concatenation is implicit (no symbol is written).
- Parentheses `()` specify precedence.
- Do not use practical regex notations such as `|` for union or `+` for repetition.

Your task is to minimize a given formal regex over the fixed alphabet {`a`, `b`, `c`, `d`}. The minimized regex must be functionally equivalent to the input and have the smallest total number of symbols, where both characters (`a`, `b`, `c`, `d`) and operations (`+`, `*`, `?`), and concatenations are counted, but parentheses are not.

Enclose your final answer within `\\boxed{}`.

Below are five input-output examples:

[Example 1]
Input Regex: **$example1_input_regex**
Output: **$example1_output**

[Example 2]
Input Regex: **$example2_input_regex**
Output: **$example2_output**

[Example 3]
Input Regex: **$example3_input_regex**
Output: **$example3_output**

[Example 4]
Input Regex: **$example4_input_regex**
Output: **$example4_output**

[Example 5]
Input Regex: **$example5_input_regex**
Output: **$example5_output**

Minimize the following regex:

Input Regex: **$regex**
Output:

---

Figure 8: The five-shot prompt used for RegexMin. It provides the formal definition of regular expressions, the task description, instructions prohibiting the use of practical regex, formatting guidelines, and includes five few-shot examples.

**RegexEq Five-shot**

You are an expert in formal regular expressions, commonly referred to as regex.

The formal regex must follow these rules:
- Allowed operations: concatenation, union (`+`), Kleene star (`*`), and option (`?`).
- Concatenation is implicit (no symbol is written).
- Parentheses `()` specify precedence.
- Do not use practical regex notations such as `|` for union or `+` for repetition.

Your task is to determine whether two given formal regexes over the fixed alphabet {`a`, `b`, `c`, `d`} are equivalent. You must output either True or False.

Enclose your final answer within `\\boxed{}`.

Below are five input-output examples:

[Example 1]
Input Regex 1: **$example1_input_regex1**
Input Regex 2: **$example1_input_regex2**
Output: **$example1_output**

[Example 2]
Input Regex 1: **$example2_input_regex1**
Input Regex 2: **$example2_input_regex2**
Output: **$example2_output**

[Example 3]
Input Regex 1: **$example3_input_regex1**
Input Regex 2: **$example3_input_regex2**
Output: **$example3_output**

[Example 4]
Input Regex 1: **$example4_input_regex1**
Input Regex 2: **$example4_input_regex2**
Output: **$example4_output**

[Example 5]
Input Regex 1: **$example5_input_regex1**
Input Regex 2: **$example5_input_regex2**
Output: **$example5_output**

Determine the equivalence of the following regexes:

Input Regex 1: **$regex1**
Input Regex 2: **$regex2**
Output:

Figure 9: The five-shot prompt used for RegexEq. It provides the formal definition of regular expressions, the task description, instructions prohibiting the use of practical regex, formatting guidelines, and includes five few-shot examples.

# H    ADDITIONAL EXPERIMENTAL RESULTS

## H.1    FULL EXPERIMENTAL RESULTS OF SECTION 5.3

This section provides the complete experimental results for all models and few-shot settings, which were not detailed in the main paper.

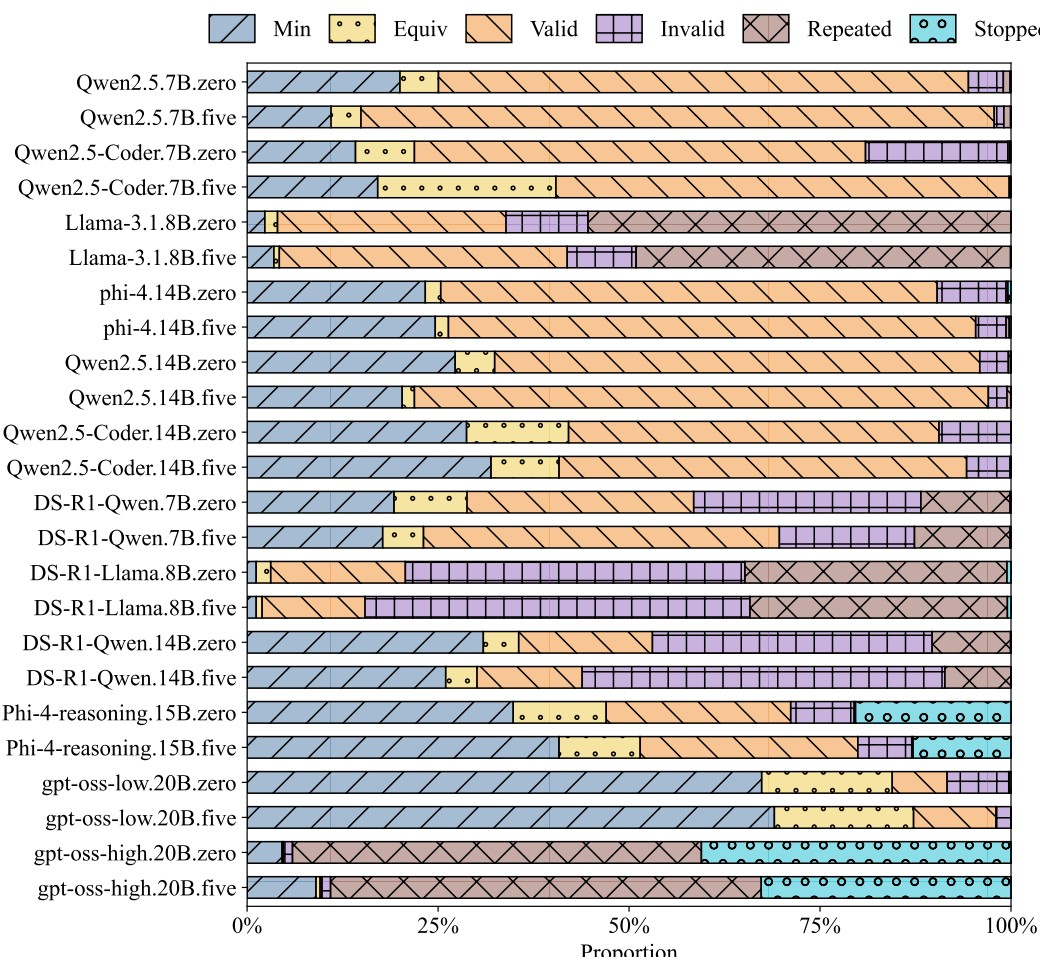

Figure 10: A bar chart of the case analysis results on RegexMin, including both zero-shot and five-shot experiments. The outcomes are categorized into the components of the confusion matrix, together with Invalid but completed answers, Repetition, and Incomplete outputs.

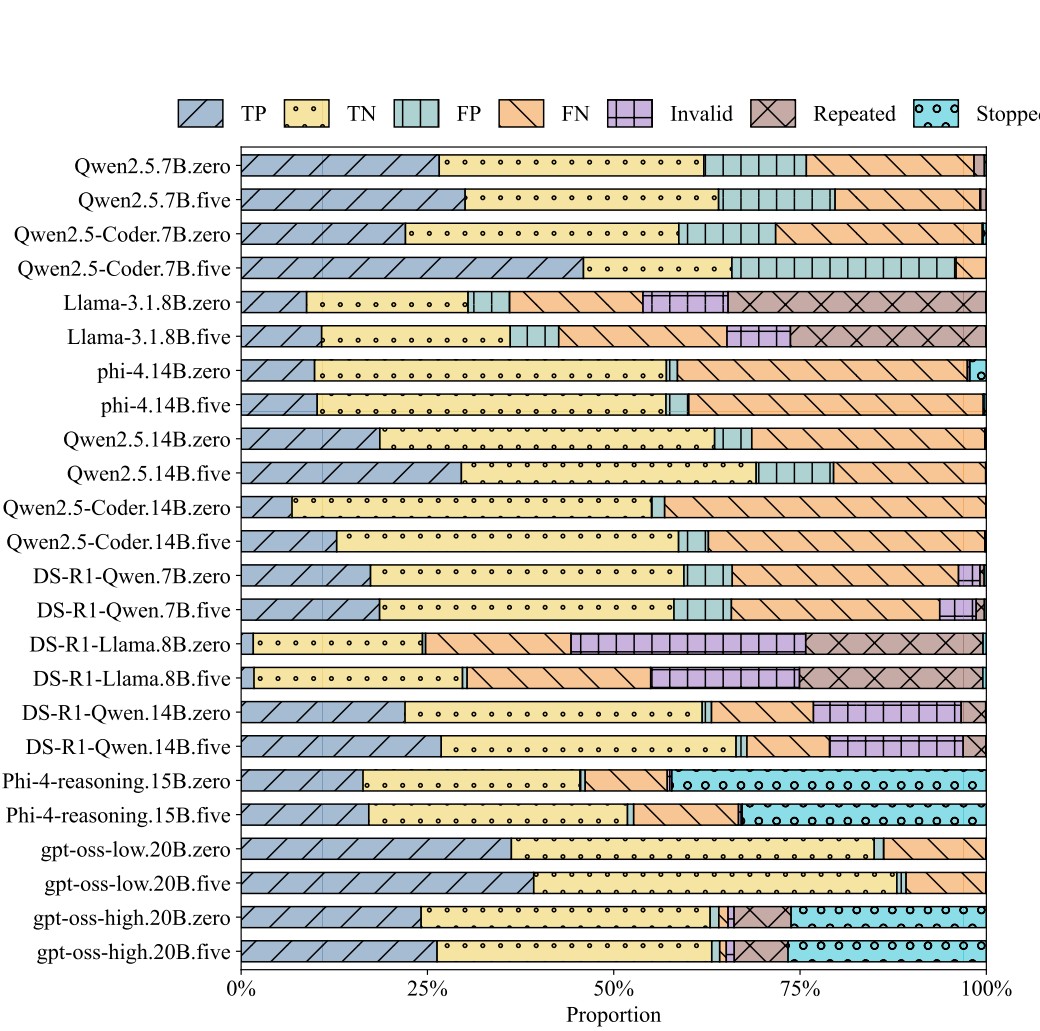

Figure 11: A bar chart of the case analysis results on RegexEq, including both zero-shot and five-shot experiments. The outcomes are categorized into the components of the confusion matrix, together with Invalid but completed answers, Repetition, and Incomplete outputs.

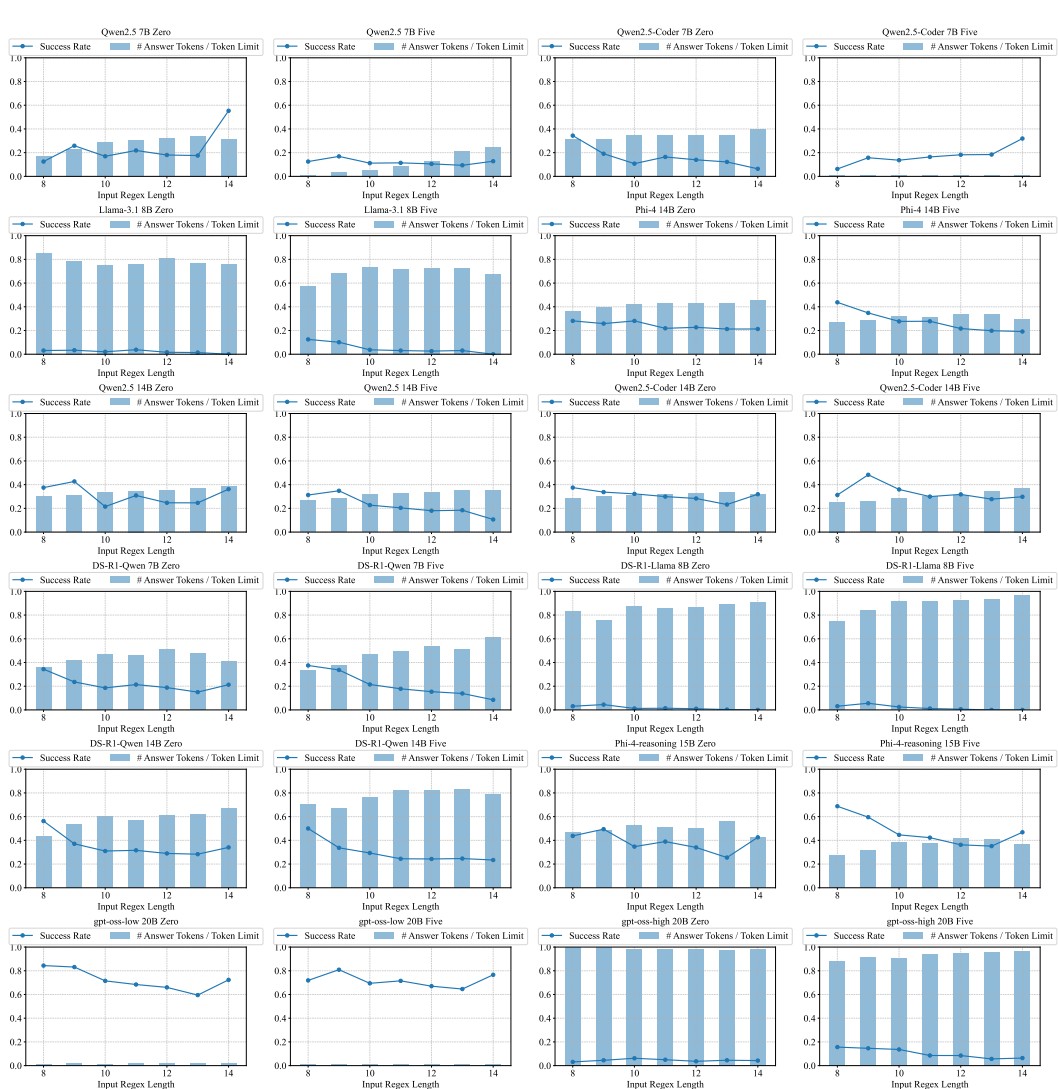

Figure 12: Final Results of Performance with Respect to Answer Length and Input Size

## H.2 Evaluation Results of 30B LLMs on RegexPSPACE

We additionally evaluate three more LLMs with 30B parameters under our evaluation settings. However, despite being recent models, their performance is unexpectedly poor, as shown in Table 12. Upon inspecting the outputs, we found that these models frequently failed to produce complete answers, often being truncated by the token limit. While analyzing the common cause of failure among these 30B models, we find that some models could decide whether to engage in reasoning via system prompts (e.g., EXAONE), or had been further tuned from such models (e.g., Qwen3-A3B). Similarly, gpt-oss also allows the extent of reasoning to be controlled via system prompts and, in high reasoning mode, it often produces overly verbose outputs that either exhausts the token budget before reaching a correct solution or degenerated into repetitive token sequences.

In order to test this hypothesis, we evaluate Qwen3-A3B on the regex minimization task in a zero-shot setting, increasing the output token limit to 4,096, as shown in Table 11. The results indicate that Qwen3-A3B does benefit from a larger token budget. However, the model still frequently fail to complete its reasoning process, often hitting the maximum token constraint before completing an answer, though without producing repeated phrases. While verbosity can sometimes help improve performance, excessive verbosity is counterproductive from a utility perspective, as it consumes unnecessary tokens without contributing meaningfully to the answer. This suggests that pretrained models should be trained in a way that mitigates such over-verbosity. Since this behavioral tendency places these models ambiguously between reasoning and non-reasoning categories, we report the corresponding experimental results solely in this section.

Table 11: Evaluation results of models with 30B parameters. Despite its large size, the models perform even worse than the 7B and 8B models.

| Model | Size | Shot | RegexMin | | | RegexEq | | |
|---|---|---|---|---|---|---|---|---|
| | | | Min. (↑) | Equi. (↑) | Ratio (↓) | Acc. (↑) | F1 (↑) | Fail (↓) |
| Qwen3-A3B | 30B | Zero | 3.56 | 3.56 | 97.81 | 6.71 | 95.71 | 92.76 |
| | | Five | 1.90 | 2.02 | 98.69 | 7.21 | 94.83 | 92.11 |
| Qwen3-Coder-A3B | 30B | Zero | 3.44 | 4.33 | 98.68 | 8.61 | 43.10 | 87.48 |
| | | Five | 6.05 | 6.82 | 97.39 | 8.81 | 44.27 | 86.85 |
| EXAONE-4.0 | 32B | Zero | 0.42 | 0.42 | 99.77 | 0.30 | 100.00 | 99.70 |
| | | Five | 9.14 | 9.20 | 96.51 | 1.07 | 83.02 | 98.66 |

Table 12: Evaluation result of Qwen3-A3B on RegexMin with different token limit. We report the proportion of responses that failed due to generating repeated tokens, the average number of tokens generated in failed responses without repetition, and the number of tokens generated in successful responses. We observe that failed responses typically produced a number of tokens close to the maximum token limit.

| Max Tokens | RegexMin | | | Fail w/ R | Fail w/o R # Tok. | Success # Tok. |
|---|---|---|---|---|---|---|
| | Min. (↑) | Equi. (↑) | Ratio (↓) | | | |
| 1024 | 3.44 | 4.33 | 98.68 | 0.12 | 1023.96 | 835.93 |
| 4096 | 17.63 | 17.80 | 91.89 | 0.89 | 4093.62 | 2441.73 |

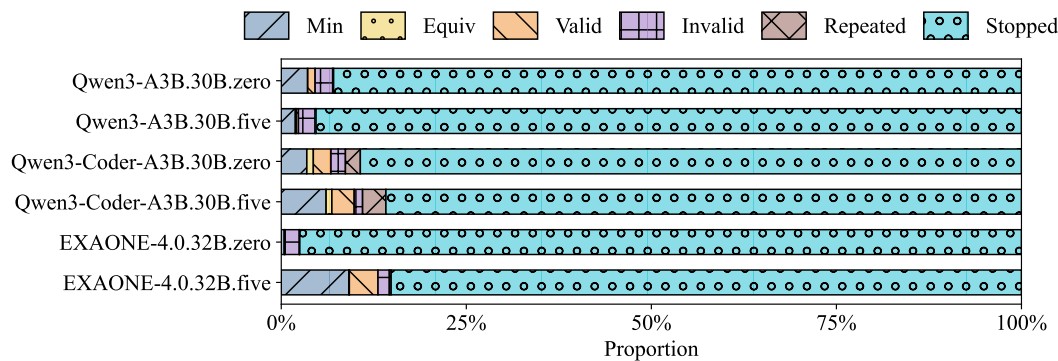

Figure 13: A bar chart of the case analysis results on RegexMin for 30B non-reasoning models.

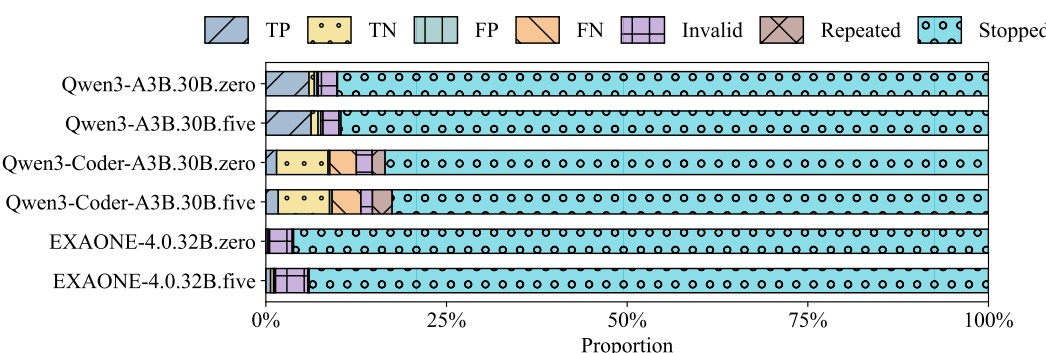

Figure 14: A bar chart of the case analysis results on RegexEq for 30B non-reasoning models.

