# OpenReview forum: "RegexPSPACE: A Benchmark for Evaluating LLM Reasoning on PSPACE-Complete Regex Problems"
_ICLR.cc/2026/Conference — ICLR 2026 Conference Withdrawn Submission_

### Official Review · Reviewer_MA8R · 2025-10-24

**Soundness:** 2
**Presentation:** 1
**Contribution:** 2
**Rating:** 2
**Confidence:** 3

**Summary:**

In this paper, the authors investigate the ability of LLMs to solve algorithmic problems in the complexity class PSPACE, in particular, the two PSPACE-complete problems **regex equivalence** (i.e., determining if two regexes are equal) and **regex minimization** (i.e., deciding the minimal size of a regex). They motivate their investigation in the beginning by claiming that "LLMs are often claimed to be Turing-complete, yet such arguments rely on unrealistic assumptons, such as infinite context length.."  and that while some recent studies investigate NP-hard problems, "the actual limitations of LLMs are more closely tied to context length, which represents limited memory, and from this spatial perspective, analyses of their computational boundaries remain underexplored". They claim that PSPACE problems are well suited to study LLMs from this "spatial perspective" and that regexes are a natural application to look at given their ubuquity throughout computer science.

First, it doesn't seem to be the case LLMs are *often* claimed to by Turing-complete and that "anlyses of their computational boundaries remain underexplored". In fact, there is a rich contemporary literature on the computational expressivity of the transformer architecture, as noted in the studies below, where it is generally understand that transformers theoretically express a comparatively inexpressive class of functions, certainly well below PSPACE (I would ask the authors to carefully consider this work and consider citing it):
> [**Merrill and Sabharwal 2025**] *Exact Expressive Power of Transformers with Padding*.  (**see citations within**)
> [**Merrill and Sabharwal 2025**] *A little depth goes a long way: the expressive power of log-depth transformers*
> [**Li and Cotterell 2025**] *Characterizing the Expressivity of Transformer Language Models*

While the focus of this paper is on empirically testing transformers on PSPACE proboems, which is tangential to the theoretical studies cited above, they also claim that "there is no direct prior work" that they can compare to and "no labeled datasets of benchmarks for PSPACE and beyond", which is not quite the case. For example, the paper below builds testing datasets for problems in the complexity classes NLOGSPACE to NEXPTIME:
> [**Madusanka, Pratt-Hartmann, Batista-navarro 2023**] *Identifying the limits of transformers when performing model-checking
with natural language*
and is worth citing (see some prior work along these things cited inside).

Given the high computational overhead of creating data instances for both problems, the dataset construction procedure for both subsets is not straightforward, and most details are included in the appendix (Appendix B). The way that this is written is the main text is hard to follow, e.g., they write (line 205) "We construct three resources for regex problems: a labeled dataset, an unlabeled dataset and a benchmark" (it remains unclear exactly what the unlabeled dataset is for). Later they write that they "Based on this [i.e., their regex dataset, we augment this dataset with a regex equivalence task] and curate 1,685 non-trivial regex problems ... to construct the Regex Problem benchmark". The details of what is being shown on the far left of Figure 2 is hard to discern, I would ask the authors to make the dataset size and structure more explicit in a table and provide an easier to read summary of the the dataset construction procedure and formal results in the appendix.

They experiment with 6 LLMs and 5 LRMs (exclusively open weight models such as Qwen, Phi, Deepseek and gpt-oss) and their main results are reported in Table 1. Specifically, they compare zero-shot versus five-shot performance and find (as expected) large gradations in performance based on model size, with GPT-oss achieving the highest accuracy of ~87% on the regex equivalence task (the results on the minimization task are harder to interpret, given that a special metric is needed which is hard to understand based on the description starting on line 248). They perform further error analysis at the end.

**Strengths:**

-- A new suite of hard reasoning challenge datasets, specifically focus on PSPACE-complete algorithmic problems involving regex manipulation.

**Weaknesses:**

-- **Essential details are difficult to follow**. Specifically, the dataset construction procedure is very hard to understand, especially given that virtually all of the details are in the appendix. I would encourage the authors to move some of this content into the main paper and show clearly the size and the structure of the two datasets in a table.

-- As discussed above, the **main justification** for why it is useful to study PSPACE problems, or other problems beyond NPTIME, needs to be better motivated and considered in light of the missing citations provided above.

-- The **choice of models is not very well motivated* and the results are difficut to comprehend (see specific questions below).

**Questions:**

-- How big are the resulting two datasets, and what is the role on the unlabeled examples you discuss at the begining of Section 4?

-- What are the 3D plots in Figure 2 showing exactly?

-- Are the resulting instances you construct computationally hard in any formal sense? I'm particularly struct by the high performance of GPT-oss-low on regex equivalence (86% acc.). One might naively thing that this shows evidence that the model can solve PTIME-complete problems, however, a more likely explanation is that the examples sampled are cover the simplest problems in that class.

Are there any known phase-transitions for this problem that you might consider exploiting in order to get at the really hard cases? This is how hard instances are constructed for NP-complete problems in the following papers, which warrants some consideration:
> [**Richardson, Sabharwal**] *Pushing the Limits of Rule Reasoning in Transformers through Natural Language Satisfiability*  [**Madusanka, Pratt-Hartmann, Batista-navarro 2023**] *Identifying the limits of transformers when performing model-checking
with natural language*

---

### Official Review · Reviewer_3xY6 · 2025-11-01

**Soundness:** 2
**Presentation:** 1
**Contribution:** 2
**Rating:** 2
**Confidence:** 3

**Summary:**

This paper proposes RegexPSPACE, a benchmark focused on PSPACE problems that includes regex equivalence and regex minimization tasks to test LLMs’ reasoning under space constraints. The authors report empirical findings from evaluating LLMs and LRMs on their benchmark: (1) models tend to perform worse on regex minimization than on equivalence; (2) common failure modes include verbosity, invalid outputs, and hitting token limits; and (3) performance drops as inputs grow longer.

**Strengths:**

- The authors evaluated their benchmark on 12 models and conducted a thorough failure-case analysis, reporting the percentage of each failure type, including repetition, invalid outputs, and stopping due to token limits.
- The evaluation metrics chosen for the two tasks are reasonable.

**Weaknesses:**

- I’m mainly concerned that the introduction motivates a benchmark to demonstrate the spatial computational limitations of LLMs (as opposed to existing theoretical work), yet the empirical findings do not clearly answer this question or convincingly achieve the goal of using PSPACE problems to quantify the need for more space complexity. The failure modes for RegexMin—such as verbosity and repetition—may relate to space pressure, but they are not unique to it. Likewise, hitting the output limit reflects an insufficient output budget, which is not necessarily caused by the spatial complexity of the benchmark’s problems. Without a causal demonstration, the claim that empirical findings necessitate evaluating "under spatial complexity constraints” is not very convincing; instead, the results indicate that longer inputs and limited output budgets correlate with worse outcomes, which is not surprising. I did a very quick literature review and could find quite a few papers with experiments showing "context length hurting the performance of LLMs" or "diminishing performance with longer prompts".
- There are quite a few typos throughout the paper. While typos are not major issues, they were noticeable during reading. A more thorough proofreading would help in a future version (including notation consistency, e.g., RegexMin vs. RegexMinimization). Some examples:
   - line 45: “a challenging and essential [task]”
   - line 83: missing a period
   - line 229: "process" misspelled
   - line 465: "focus" misspelled

**Questions:**

- Is there a way to explicitly quantify how much the implicit space requirements of these PSPACE tasks contribute to LLM failures?
- Beyond the reported metrics, are there any novel insights or qualitative analyses that emerge from the empirical results of evaluating this benchmark?

---

### Official Review · Reviewer_KbuW · 2025-11-01

**Soundness:** 2
**Presentation:** 3
**Contribution:** 2
**Rating:** 4
**Confidence:** 5

**Summary:**

this paper proposes a complexity-based benchmark based on PSPACE complexity, extending beyond traditional NP-complexity complexity hierarchy

**Strengths:**

1. introduce PSPACE in evaluation, which is not included or considered in any previous benchmarks
2. solid experiment results with open-source models
3. detailed failure case analysis

**Weaknesses:**

1. there are plenty works trying to evaluate LLMs based on complexity hierarchy, as discussed related work, and thus I am not sure why adding this P-SPACE complexity class experiment is essentially interesting. What's inherently new about it and what different observations can we derive compared with eval results using time-complexity classes?
2. the paper only evaluates open-source models without close-source models

**Questions:**

why PSPACE? a lot of models can't work well on NP-complete problems yet, why we need this extra benchmark?

What new observations or conclusions can be made by evaluating on PSPACE that cannot be found when evaluating on NP-complete etc?

---

### Author Response · Authors · 2025-11-19
**General Response**

We would like to thank all reviewers for their constructive comments.

However, there seem a gap in understanding the fundamental necessity of our problem's PSPACE-completeness or imprecise feedbacks on related works (e.g.; the paper “Identifying the limits of transformers...” addresses a PTIME problem, representing a significantly lower complexity bound than our PSPACE-complete task .) due to various reasons.

We will revise our paper to cope with these comments and clarify our contributions compared with current works more clearly.

---

### Note · Authors · 2025-11-19

I have read and agree with the venue's withdrawal policy on behalf of myself and my co-authors.